# Evidence of off-target probe binding affecting 10x Genomics Xenium gene panels compromise accuracy of spatial transcriptomic profiling

Caleb Hallinan[1,2], Hyun Joo Ji[1,3], Edmund Tsou[1,2,3], Steven L Salzberg[1,2,3,4], Jean Fan[1,2,3]*

[1]Center for Computational Biology, Whiting School of Engineering, Johns Hopkins University, Baltimore, United States; [2]Department of Biomedical Engineering, Johns Hopkins University, Baltimore, United States; [3]Department of Computer Science, Johns Hopkins University, Baltimore, United States; [4]Department of Biostatistics, Johns Hopkins University, Baltimore, United States

*For correspondence:
jeanfan@jhu.edu

Competing interest: The authors declare that no competing interests exist.

## eLife Assessment

This **valuable** study identifies and characterizes probe binding errors in a widely used commercial platform for spatial transcriptomics, discovering that at least 14 out of 313 genes in a human breast cancer panel are not accurately detected. The authors provide **convincing** evidence for their findings through validation against multiple independent sequencing technologies and reference datasets, and they introduce a computational tool to help predict potential off-target probe binding. Given the broad adoption of this platform in biomedical research, this work provides an essential quality control resource that will improve data interpretation across numerous studies.

**Abstract** The accuracy of spatial gene expression profiles generated by probe-based in situ spatially resolved transcriptomic technologies depends on the specificity with which probes bind to their intended target gene. Off-target binding, defined as a probe binding to something other than the target gene, can distort a gene's true expression profile, making probe specificity essential for reliable transcriptomics. Here, we investigated off-target binding affecting the 10x Genomics Xenium technology. We developed a software tool, Off-target Probe Tracker (OPT), to identify putative off-target binding via alignment of probe target sequences and assessing whether mapped loci corresponded to the intended target gene across multiple reference annotations. Applying OPT to a Xenium human breast gene panel, we identified at least 14 out of the 313 genes in the panel potentially impacted by off-target binding to protein-coding genes. To substantiate our predictions, we leveraged a Xenium breast cancer dataset generated using this gene panel and compared results to orthogonal spatial and single-cell transcriptomic profiles from Visium CytAssist and 3′ single-cell RNA-seq derived from the same tumor block. Our findings indicate that for some genes, the expression patterns detected by Xenium demonstrably reflect the aggregate expression of the target and predicted off-target genes based on Visium and single-cell RNA-seq, rather than the target gene alone. We further applied OPT to identify potential off-target binding in custom gene panels and integrate tissue-specific RNA-seq data to assess effects. Overall, this work enhances the biological interpretability of spatial transcriptomics data and improves reproducibility in spatial transcriptomics research.

**Figure 1.** Schematic of potential off-target binding in 10x Genomics Xenium. In this illustration, the arms of the padlock probes were designed to bind an RNA sequence intended to correspond to a target gene (green). However, these probes exhibit off-target binding and bind to an RNA sequence in a different off-target gene (red). The probe is circularized and subsequently amplified via rolling circle amplification (RCA). Hybridization of fluorescent probes to the RCA product enables the generation of a fluorescent signal that is used to quantify RNA expression within cells.

## Introduction

Recent advances in high-throughput spatially resolved transcriptomic profiling technologies have enabled the investigation of gene expression with high spatial resolution within tissues. One such commercially available spatial transcriptomics platform is Xenium from 10x Genomics, a publicly traded company with a market capitalization exceeding $2 billion as of November 2025 (*Yahoo Finance, 2025*). Xenium achieves spatial gene expression profiling at single-cell resolution for targeted genes using a probe-based in situ detection approach. 10x Genomics currently offers targeted gene panels with pre-designed probe sets. As of December 2024, over 16,000 Xenium consumable reactions have been sold, with each tissue slide profiled costing approximately $5000, underscoring the platform's widespread use and high commercial value (*10x Genomics, 2025a*; *10x Genomics, 2025b*).

Briefly, Xenium uses padlock probes that include sequences complementary to the RNA of target genes. Once a padlock probe binds to its target, it is ligated and subsequently amplified via rolling circle amplification (RCA). Fluorescently labeled decoder probes then hybridize to the amplified RCA product, enabling the simultaneous detection and decoding of the optical signature, or codeword, specific to each gene in the panel through successive rounds of fluorescence imaging. When combined with cell segmentation, this approach allows for spatially resolved single-cell quantification of gene expression.

The accuracy of these gene expression measurements thus relies on the specificity of the probes to bind to their intended target gene. We define off-target binding as when a probe binds to something other than the RNA sequence intended to correspond to the target gene (*Figure 1*). We note once ligation and RCA occur, the resulting fluorescent signal cannot be distinguished between on- and off-target binding. As such, off-target binding can distort the quantification of the intended target gene's expression, as the observed expression would represent a combination of the target as well as off-target expression.

To predict for such potential off-target binding, we developed Off-target Probe Tracker (OPT), a software tool that aligns probe target sequences to an annotated transcriptome with the option to allow for mismatches that may still permit probe binding. Using OPT, we identify putative off-target probe binding to protein-coding genes affecting at least 14 out of 313 genes in a 10x Genomics Xenium human breast gene panel, compromising the accuracy of their spatial transcriptomic profiles. We substantiate our predictions using data from orthogonal spatial and single-cell gene expression profiling technologies. We further apply OPT to identify potential off-target binding in custom gene panels and integrate tissue-specific RNA-seq data from the Human BioMolecular Atlas Program (HuBMAP) to assess whether such off-target binding could meaningfully affect assayed expression patterns in specific tissues. By facilitating a more rigorous evaluation of probe specificity, tools like OPT can aid in future probe design decisions to help ensure that probes are optimized to minimize off-target binding based on current transcriptome annotations.

## Results

### OPT predicts potential off-target probe binding

To identify potential off-target binding impacting the 10x Genomics Xenium technology, we require the probe target sequences for a specific gene panel of interest, generally represented in a FASTA file. To this end, we first focus on a human breast gene panel used in the Janesick et al. publication, courtesy of 10x Genomics (Methods; *Supplementary file 1*). This file includes 2582 probe target sequences that are 40 bp in length and designed to target 313 genes, including 33 genes targeted by custom probes, with an average of 8 probe target sequences per gene (ranging from 2 to 21 probe target sequences per gene). We note this panel represents an earlier iteration of and is highly similar to the commercially available pre-designed Xenium v1 Human Breast Gene Expression Panel (Appendix Note).

To enable the prediction of potential off-target binding, we developed a software tool called OPT (Methods) that uses nucmer (*Marçais et al., 2018*) to align probe target sequences to various reference transcriptomes, which comprise curated collections of transcript isoforms for all genes in a species. OPT features adjustable parameters for binding strictness (e.g., number of mismatches) and generates a summary file that details all targeted genes along with their potential off-targets based on the sequence alignments. 10x Genomics designed its probe target sequences using the GENCODE 'basic' annotation (*Mudge et al., 2025*), so initially we also used the latest GENCODE 'basic' annotation (v47) to predict off-target binding for these probes.

We first sought to predict if a probe has off-target binding based on perfect sequence homology (i.e., if it aligns with 100% identity) with any annotated transcripts other than those that belong to the intended target gene. Of the 2582 probe target sequences in this gene panel, using GENCODE v47, OPT identified 121 probe target sequences across 37 genes as having off-target binding based on perfect sequence homology (*Table 1*). Among the 37 genes with predicted off-target binding, the number of affected probe target sequences per gene ranged from 1 to 8. Overall, these off-target probes matched 71 other genes, including 20 protein-coding genes, 31 pseudogenes, 10 long non-coding RNAs, 9 transcripts labeled as nonsense-mediated decay, and 1 microRNA gene.

### Off-target binding predictions vary across different annotations

OPT relies on alignments to an annotated transcriptome, which ideally reflects all genes and gene variants stably transcribed in a given species. However, genome annotation is still an active area of research (*Varabyou et al., 2023*), with discrepancies across annotations in gene counts, isoforms, and many other features. We therefore further used OPT to predict for off-target binding and affected genes using two additional human genome annotation sets, RefSeq (v110), (*O'Leary et al., 2016*) and CHESS (v3.1.3) (*Varabyou et al., 2023*), and compared to our previous results from GENCODE (v47) (*Mudge et al., 2025*) (Methods).

When considering only perfect sequence homology, while we previously found 37 affected genes using GENCODE, we found 14 when using RefSeq and 23 when using CHESS (*Supplementary files 2 and 3*). Given that RefSeq and CHESS have more transcripts than GENCODE, these discrepancies in off-target binding predictions was not simply an artifact of the difference in transcript set sizes.

While the human annotation databases mostly agree on the number of protein-coding genes in the genome, they remain widely divergent on pseudogenes and lncRNA genes. Therefore, we focused on how the results change when we restrict our analysis to only protein-coding genes. By excluding pseudogenes (which are presumably not expressed), lncRNAs, and other non-protein-coding RNAs when using OPT, the number of affected genes fell to 11 for GENCODE, 10 for RefSeq, and 9 for CHESS (*Supplementary file 4*). Again, these discrepancies reflect annotation differences. For example, the probe target sequence (ENSG00000196154|S100A4|ab4e3dc), which was designed to target *S100A4* based on GENCODE annotations, also aligns to *S100A5* in RefSeq (*Appendix 1—figure 1*). We reason that if a probe target sequence aligns off-target to a protein-coding gene based on any of these annotations, it could result in off-target binding. We therefore focused further analysis on the union of genes with predicted off-target binding to protein-coding genes across the 3 annotations, resulting in 14 genes: *ADH1B, AKR1C1, APOBEC3A, APOBEC3B, AQP1, C1QA, CD79B, CD8B, CEACAM6, POLR2J3, S100A4, TOMM7, TPD52,* and *TPSAB1* (*Appendix 1—figure 2*; *Supplementary file 5*).

**Table 1.** Off-target Probe Tracker (OPT) output of genes with predicted off-target binding based on perfect sequence homology using GENCODE v47.

This table shows the 37 genes whose probes in the 10x Genomics Xenium v1 Human Breast Gene Expression Panel exhibit predicted off-target probe binding, where each off-target alignment involves a perfect 40 bp match to the probe target sequence. Although OPT predicted off-target binding of CCPG1 probe target sequences to the DNAAF1-CCPG1 gene, we manually excluded it from our list because DNAAF1-CCPG1 is a read-through gene containing portions of both DNAAF1 and CCPG1. The final column shows the gene types, in order, of each of the off-target genes shown in column 3. Abbreviations: PC = protein-coding; PG = pseudogene; NMD = nonsense-mediated decay; lncRNA = long non-coding RNA.

| Target gene | Number of probes | Predicted binding genes | Number of probes aligned | Gene types – GENCODE (v47) |
|---|---|---|---|---|
| ADH1B | 8 | ADH1B, ADH1A, ADH1C | 8, 2, 1 | PC, PC, PC |
| AKR1C1 | 9 | AKR1C1, AKR1C2, AKR1C3, AKR1C4, AKR1C5P | 9, 1, 1, 1, 1 | PC, PC, PC, PC, PG |
| APOBEC3A | 8 | APOBEC3A, APOBEC3B | 8, 2 | PC, PC |
| APOBEC3B | 8 | APOBEC3B, APOBEC3D, APOBEC3F, ENSG00000284554 | 8, 2, 2, 2 | PC, PC, PC, PC |
| AQP1 | 10 | AQP1, ENSG00000250424 | 10, 4 | PC, PC |
| C15orf48 | 6 | C15orf48, MIR147B | 6, 1 | PC, miRNA |
| C1QA | 4 | C1QA, ENSG00000289692 | 4, 2 | PC, PC |
| CD68 | 7 | CD68, ENSG00000264772 | 7, 6 | PC, lncRNA |
| CD79B | 5 | CD79B, ENSG00000285947 | 5, 3 | PC, PC |
| CD8B | 16 | CD8B, CD8B2 | 16, 2 | PC, PC |
| CEACAM6 | 8 | CEACAM6, ENSG00000267881 | 8, 1 | PC, PC |
| CLECL1; CLECL1P | 3 | CLECL1P, ENSG00000293488 | 3, 3 | PG, lncRNA |
| DPT | 8 | DPT, LINC00970 | 8, 8 | PC, lncRNA |
| EPCAM | 8 | EPCAM, ENSG00000225356 | 8, 1 | PC, PG |
| HMGA1 | 7 | HMGA1, HMGA1P1, HMGA1P2, HMGA1P3 | 7, 1, 1, 1 | PC, PG, PG, PG |
| IL2RG | 9 | IL2RG, ENSG00000285171 | 9, 8 | PC, NMD |
| KRT14 | 6 | KRT14, KRT16P6, ENSG00000290977 | 6, 1, 1 | PC, PG, lncRNA |
| KRT8 | 16 | KRT8, KRT8P3, KRT8P2, KRT8P33, KRT8P45, CDK5R2-AS1, ENSG00000304440, KRT8P11, KRT8P17, KRT8P22, KRT8P30, KRT8P32, KRT8P36, KRT8P37, KRT8P42 | 16, 3, 2, 2, 2, 1, 1, 1, 1, 1, 1, 1, 1, 1, 1 | PC, PG, PG, PG, PG, lncRNA, lncRNA, PG, PG, PG, PG, PG, PG, PG, PG |
| LDHB | 8 | LDHB, ENSG000002854 | 8, 5 | PC, NMD |
| LILRA4 | 8 | LILRA4, ENSG00000275210 | 8, 1 | PC, lncRNA |
| MYLK | 11 | MYLK, MYLKP1 | 11, 1 | PC, PG |
| MYO5B | 8 | MYO5B, MYO5BP1, MYO5BP2, ENSG00000266997 | 8, 1, 1, 4 | PC, PG, PG, NMD |
| PCLAF | 8 | PCLAF, ENSG00000259316 | 8, 1 | PC, NMD |
| POLR2J3 | 10 | POLR2J3, POLR2J4, POLR2J, ENSG00000270249, POLR2J2, POLR2J2-UPK3BL1, ENSG00000291154 | 10, 4, 3, 2, 2, 2, 1 | PC, lncRNA, PG, PC, PC, PC, NMD, lncRNA |
| PTGDS | 5 | PTGDS, ENSG00000284341 | 5, 3 | PC, NMD |
| SCD | 8 | SCD, SCDP1 | 8, 2 | PC, PG |
| SERHL2 | 8 | SERHL2, SERHL | 8, 7 | PC, PG |
| SERPINA3 | 8 | SERPINA3, ENSG00000273259 | 8, 8 | PC, NMD |
| SLAMF1 | 10 | SLAMF1, ENSG00000228863 | 10, 1 | PC, lncRNA |

*Table 1 continued on next page*

Table 1 continued

| Target gene | Number of probes | Predicted binding genes | Number of probes aligned | Gene types – GENCODE (v47) |
|---|---|---|---|---|
| SMS | 8 | SMS, ENSG00000213080, ENSG00000232389, ENSG00000249779 | 8, 3, 1, 1 | PC, PG, PG, PG |
| THAP2 | 13 | THAP2, ENSG00000258064 | 13, 2 | PC, NMD |
| TPD52 | 8 | TPD52, ENSG00000276418 | 8, 5 | PC, NMD |
| TPSAB1 | 2 | TPSAB1, TPSB2, TPSD1 | 2, 2, 1 | PC, PC, PC |
| TRAF4 | 9 | TRAF4, ENSG00000225869 | 9, 1 | PC, PG |
| TUBB2B | 8 | TUBB2B, TUBB2BP1 | 8, 1 | PC, PG |
| VOPP1 | 11 | VOPP1, ENSG00000223612 | 11, 1 | PC, PG |
| VWF | 8 | VWF, VWP1 | 8, 1 | PC, PG |

## Comparison with Visium CytAssist reveals spatial gene expression patterns consistent with off-target binding

To investigate the potential effects of our predicted off-target binding for this Xenium human breast gene panel in experimental settings, we compared spatial gene expression patterns detected in two previously published spatial transcriptomics datasets from serial sections of the same breast cancer tissue: one section assayed with Xenium using this gene panel, and another assayed using Visium CytAssist, an orthogonal spatial transcriptomics platform (*Janesick et al., 2023*). Briefly, Visium CytAssist is a sequencing-based spatial transcriptomics technology in which RNA is hybridized to spatially barcoded capture spots on a slide, enabling spatial transcriptomic mapping after sequencing. However, while Xenium offers single-cell resolution gene expression quantification, Visium quantifies gene expression within 55 µm spots. To enable direct comparison, we first structurally aligned the Xenium and Visium tissue sections using STalign (*Clifton et al., 2023*), restricting our analysis to over-lapping regions since different parts of the tissue were profiled (*Appendix 1—figure 3A*). To improve visual comparability, we aggregated the Xenium gene expression data at the aligned locations to match the Visium spatial resolution and visualized using resolution-matched tiles (*Appendix 1—figure 3B*; Methods).

Among the 14 genes exhibiting off-target binding to protein-coding genes, 4 were present in the Visium dataset that had at least one corresponding off-target gene also detected in the dataset. For genes with no predicted off-target binding based on perfect sequence homology such as *MS4A1*, we observed a visually similar spatial pattern between the two technologies (*Figure 2A*), suggesting that spatially aligned groups of cells across the two technologies express this gene at comparable relative magnitudes. This can be quantitively assessed by comparing the pseudo-log gene expression values from both Visium and Xenium at matched spatial locations and computing the root-mean-square error (RMSE) and Pearson correlation in a manner similar to STcompare (*Clifton et al., 2025*). For *MS4A1*, the RMSE is relatively low at 3.746, and the Pearson correlation of 0.382 indicates a moderate degree of concordance between the two technologies. However, for genes with predicted off-target binding based on perfect sequence homology such as *APOBEC3B*, we observed a visually dissimilar spatial pattern between the two technologies (*Figure 2B*). Consistent with this, the RMSE is relatively high at 5.452, and the Pearson correlation is *nan* because *APOBEC3B* is not expressed in the Visium dataset. Importantly, its predicted off-target genes, *APOBEC3D* and *APOBEC3F*, in Visium show a visually more similar spatial pattern to the Xenium *APOBEC3B*. To better visualize the effect of off-target binding within the Xenium data, we aggregated the expression of each gene along with its predicted off-targets found in the Visium dataset and visually compared across spatial locations. Notably, the spatial pattern of the aggregated expression of *APOBEC3B*, *APOBEC3D,* and *APOBEC3F* in Visium is visually more similar to the spatial pattern of *APOBEC3B* in Xenium. Quantitatively, comparing this aggregated expression results in a decrease in RMSE to 4.465 and a non-*nan* Pearson correlation of 0.160, consistent with the prediction that Xenium *APOBEC3B* probes exhibit off-target binding to *APOBEC3D* and *APOBEC3F*. We further visually confirmed using the Integrative Genomics Viewer

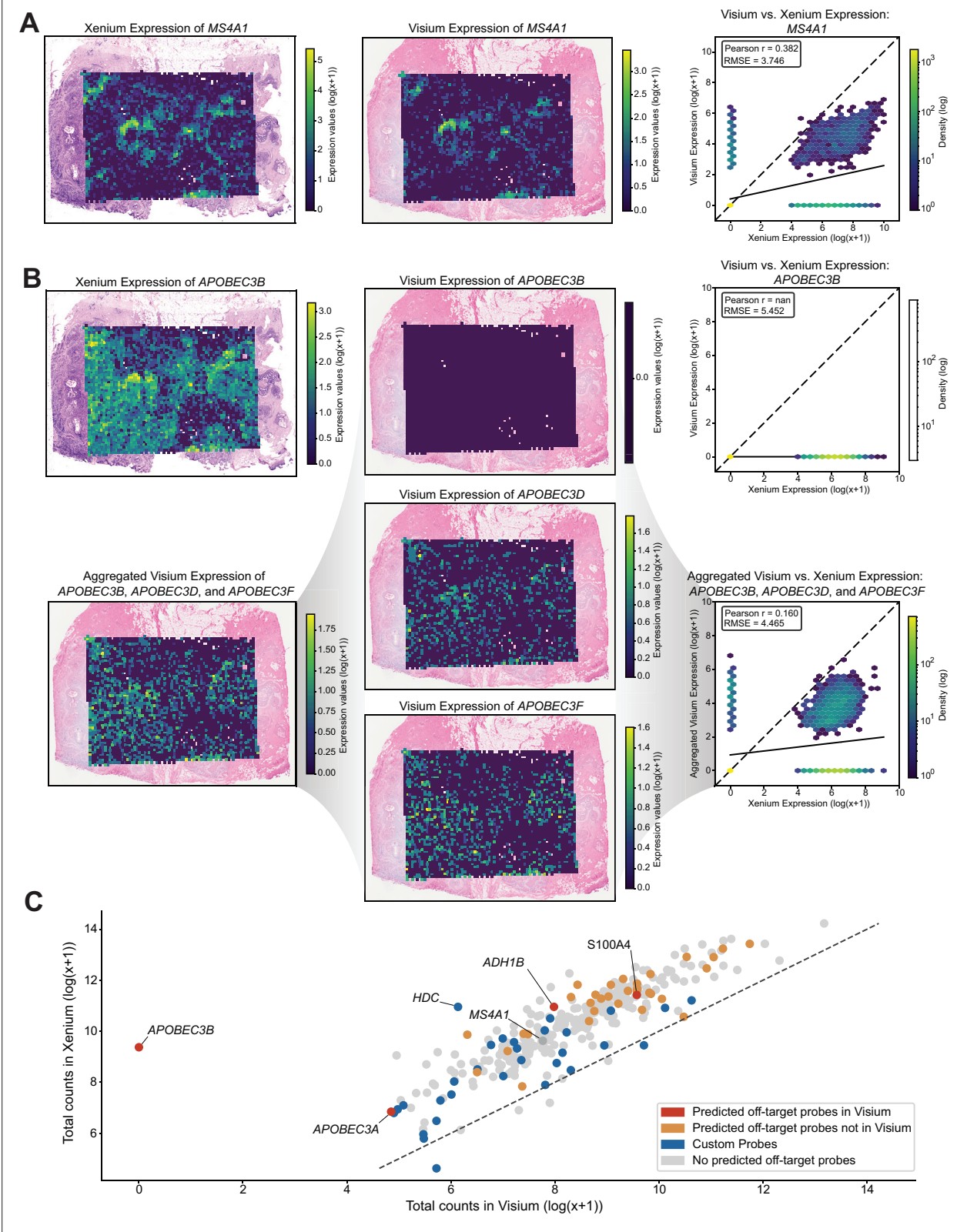

**Figure 2.** Comparison of spatial gene expression patterns between Xenium and Visium. (**A**) Spatial gene expression of MS4A1 overlaid on the corresponding histological images for Xenium and Visium, accompanied by a density plot comparing Xenium vs. Visium MS4A1 expression. The dotted line indicates the identity line (*X = Y*), and the solid line represents the line of best fit. (**B**) Gene expression patterns for APOBEC3B: Xenium expression, Visium expression, the aggregated Visium expression combining APOBEC3B and its predicted off-target gene's expression APOBEC3D and APOBEC3F,

*Figure 2 continued on next page*

Figure 2 continued

and Visium expression of APOBEC3B's predicted off-targets APOBEC3D and APOBEC3F. Two density plots are shown: one comparing Xenium vs. Visium for APOBEC3B alone, and one comparing Xenium vs. the aggregated Visium expression of APOBEC3B with all off-targets. The dotted line indicates the identity line ($X = Y$), and the solid line represents the line of best fit. (**C**) Scatterplot of log-transformed total expression counts (with a pseudocount) for 307 genes comparing Visium and Xenium data. The dotted line indicates the identity line ($X = Y$), and points (genes) are colored by probe information.

(IGV) where two probes intending to bind to *APOBEC3B* had target sequences found to perfectly align to sequences in both *APOBEC3D* and *APOBEC3F*, consistent across all annotations evaluated (*Appendix 1—figure 4*).

Overall, when comparing the total gene expression between the two technologies, we observed a generally strong positive correlation, consistent with the previously published work (*Janesick et al., 2023*). We do not observe an obvious trend between gene expression magnitude and the presence of predicted off-target probes (*Figure 2C*), suggesting that off-target binding prediction alone does not explain the observed higher expression magnitude in Xenium compared to Visium, which may still be attributed to variation in detection efficiency, sequencing depth, and other factors.

## Comparison with scRNA-seq reveals single-cell gene expression patterns consistent with off-target binding

To further investigate the potential effects of our predicted off-target binding for this Xenium human breast gene panel, we compared the detected single-cell gene expression patterns in the same previously published work using Chromium Next GEM Single Cell 3′ (*Janesick et al., 2023*). Briefly, single-cell RNA sequencing (scRNA-seq) with 3′ end capture is a technique used to profile gene expression at the single-cell level by profiling the 3′ ends of mRNA transcripts with sequencing followed by alignment to a genome or transcriptome for quantification. While this approach provides single-cell resolution gene expression quantification, it lacks spatial information. To enable a single-cell comparison with Xenium, we use Harmony (*Korsunsky et al., 2019*) to remove batch effects and project cells into a shared Uniform Manifold Approximation and Projection (UMAP) embedding (*Appendix 1—figure 5A, B*). We also performed Leiden clustering on the harmonized principal components (PCs) to quantitatively compare cluster expression (*Appendix 1—figure 5C*; Methods).

Among the 14 genes exhibiting off-target binding to protein-coding genes, 10 were present in the scRNA-seq dataset that had at least one corresponding off-target gene also detected in the dataset. Again, for genes with no predicted off-target binding based on perfect sequence homology such as *MS4A1*, we observed a visually similar gene expression pattern in the harmonized UMAP across both technologies (*Figure 3A*), suggesting that transcriptionally similar clusters of cells or cell types across the two technologies express this gene at comparable relative magnitudes. We further quantitatively assessed the data by comparing the pseudo-log gene expression values obtained from the clusters within the clustered harmonized UMAP and computed the RMSE and Pearson correlation. For *MS4A1*, the RMSE is relatively low at 0.479, and the Pearson correlation of 0.991 indicates a strong degree of concordance between the two technologies. Likewise, again, for genes with predicted off-target binding based on perfect sequence homology such as *APOBEC3B*, we observed a visually dissimilar gene expression pattern on the harmonized UMAP (*Figure 3B*). Consistent with this, the RMSE is relatively high at 0.829, and the Pearson correlation is *nan* because *APOBEC3B* is not expressed in the scRNA-seq dataset. Again, its predicted off-target genes, *APOBEC3D* and *APOBEC3F*, in scRNA-seq showed a visually more similar expression pattern in the harmonized UMAP embedding to the Xenium *APOBEC3B*. To better illustrate the impact of the off-target probes, we again aggregated the expression of a gene and its predicted off-target genes present in the scRNA-seq data and visually compared across the harmonized UMAP embedding. The aggregated expression of *APOBEC3B*, *APOBEC3D*, and *APOBEC3F* in the scRNA-seq data shows a visually more similar gene expression pattern in the harmonized UMAP embedding to *APOBEC3B* in the Xenium data (*Figure 3B*). Quantitatively, comparing this aggregated expression results in a decrease in RMSE to 0.596 and a non-nan in Pearson correlation to 0.417, consistent with the prediction that Xenium *APOBEC3B* probes exhibit off-target binding with these paralogs.

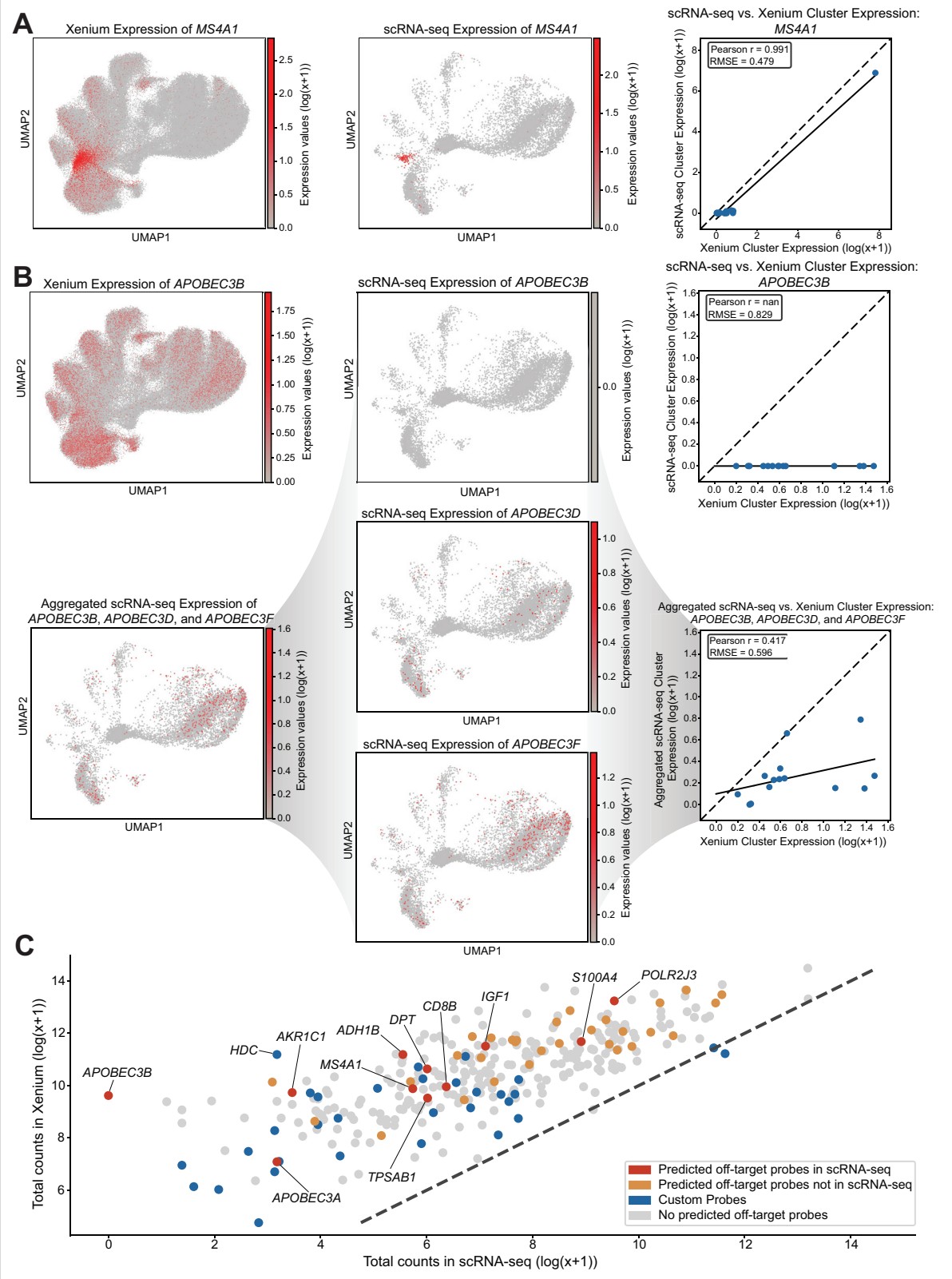

**Figure 3.** Comparison of single-cell gene expression patterns between Xenium and single-cell RNA sequencing (scRNA-seq). (**A**) Harmonized Uniform Manifold Approximation and Projection (UMAP) visualization of MS4A1 expression for Xenium and scRNA-seq data, accompanied by a scatterplot comparing Xenium vs. scRNA-seq MS4A1 cluster expression. The dotted line indicates the identity line (X = Y), and the solid line represents the line of best fit. (**B**) Comparison of APOBEC3B expression patterns on harmonized UMAP: Xenium expression, scRNA-seq expression, an aggregated

*Figure 3 continued on next page*

*Figure 3 continued*

scRNA-seq profile combining APOBEC3B and its predicted off-target genes' expression APOBEC3D and APOBEC3F, and scRNA-seq expression of APOBEC3B's predicted off-targets APOBEC3D and APOBEC3F. Two scatterplots are shown: one comparing Xenium vs. scRNA-seq for APOBEC3B cluster expression alone, and one comparing Xenium vs. the aggregated scRNA-seq cluster expression of APOBEC3B and its predicted off-targets. The dotted line indicates the identity line ($X = Y$), and the solid line represents the line of best fit. (**C**) Scatterplot of log-transformed total expression counts (with a pseudocount) for 313 genes between Visium and scRNA-seq data. The dotted line indicates the identity line ($X = Y$), and points (genes) are colored by probe information.

Overall, when comparing total gene expression between the two technologies, we again observed a generally strong positive correlation (*Figure 3C*), similar to the Visium comparison results and consistent with the previously published work (*Janesick et al., 2023*).

## OPT results when allowing mismatches at the terminal ends of the probe target sequences identify additional off-target candidates

Thus far, we have focused on predicting off-target binding based on perfect sequence homology. However, we reason that imperfect sequence matching could still result in off-target binding. Specifically, for Xenium v1, which underlies the Xenium human breast gene panel, padlock probes with two 20 base pair (bp) arms bind complementary mRNA regions, forming a 40-bp probe target sequence. A ligase then circularizes the padlock probe, favoring specific 2 bp junctions. Importantly, if there is a sequence mismatch, particularly outside the ligation site toward the terminal ends of the probe target sequence, hybridization may still occur and result in off-target binding, albeit with reduced hybridization efficiency (*Appendix 1—figure 6A*). We therefore added an option in OPT to allow imperfect alignments at the ends of the probe target sequences, specifying the sequence length at either end, where mismatches, insertions, deletions, or clipping can occur (Methods).

Allowing for 10 bp mismatches on either end of the 40 bp probe target sequence (i.e., requiring a 20-bp match covering the middle of the probe target sequence including the ligation site) revealed 18 additional genes, including protein-coding genes, with potential off-target binding when using GENCODE v47 (*Supplementary file 6*), among which *ACTG2* was included. Additionally, 10 of the 37 genes previously predicted to be affected by off-target binding based on perfect sequence homology were now predicted to have additional off-target genes, including protein-coding genes (*Supplementary file 7*), among which *TUBB2B* was included. Both genes showed visually dissimilar spatial patterns between Xenium and Visium, accompanied by comparably high RMSE and low Pearson correlation (*Appendix 1—figure 7A, B*). In contrast, the spatial pattern of the aggregate of *TUBB2B* and *ACTG2* with their predicted off-target protein-coding genes (*TUBB2A* and *ACTB/ACTA1/POTEM*, respectively) in Visium more closely resembled the spatial pattern of *TUBB2B* and *ACTG2* observed in Xenium, accompanied by a corresponding decrease in RMSE for both genes and an increase in Pearson correlation for *TUBB2B*. Likewise, a similar trend is observed in the scRNA-seq comparison (*Appendix 1—figure 8A, B*). Ultimately, these findings suggest that off-target binding, even with imperfect sequence matching, can contribute to the expression patterns observed in Xenium.

## RNA-seq reference atlases suggest off-target binding can variably impact results in Xenium custom probe panels

Our analyses to this point focused on the Xenium human breast gene panel from Janesick et al., a predecessor to and thus similar to the commercially available pre-designed Xenium v1 Human Breast Gene Expression Panel (Appendix Note). We next sought to determine whether such off-target binding could affect custom Xenium gene panels, in which gene selection is specified by the user rather than provided as pre-designed panels by 10x Genomics. To investigate this, we leveraged two custom Xenium gene panels used by the HuBMAP (*Jain et al., 2023*): one designed for the placenta and another designed for the kidney, lung, and heart (i.e., multi-organ). Applying OPT to the associated FASTA files using the GENCODE v47 annotation and allowing for 10 bp mismatches on either end of the 40 bp probes, we found that 49 genes out of the 300 targeted genes in the placenta panel (*Supplementary file 8*) and 24 genes out of the 300 targeted genes in the multi-organ (kidney, lung, and heart) panel had predicted off-target binding (*Supplementary file 9*). Of these, 30 of the 49 placenta panel genes and 11 of the 24 multi-organ panel genes had predicted off-targets that were protein-coding genes.

To assess the potential effects of our predicted off-target binding for these custom Xenium gene panels in experimental settings, we examined the expression of the corresponding predicted off-target genes in matched scRNA-seq or bulk RNA-seq data from the HuBMAP consortium for each of the relevant tissue types. For the placenta custom panel, 34 of the 49 genes with predicted off-target genes were detected with non-zero expression magnitudes in the placenta bulk RNA-seq dataset (*Appendix 1—figure 9*). For the multi-organ panel, of the 21 genes with predicted off-target genes, 13 were detected with non-zero expression magnitudes in the heart, 12 in the kidney, and 13 in the lung (*Appendix 1—figure 10A–C*). Together, these results suggest that off-target binding can impact custom Xenium gene panels by distorting observed Xenium gene expression measurements in a tissue-dependent manner, particularly where the off-target gene is expressed at a higher magnitude compared to the target gene within the tissue of interest.

## Discussion

Our study presents evidence of off-target probe binding that may distort gene expression profiles affecting the 10x Genomics Xenium spatial transcriptomics technology. We identified at least 14 out of the 313 genes in a Xenium human breast gene panel, which is highly similar to the commercially available pre-designed Xenium v1 Human Breast Gene Expression Panel (Appendix Note), that may be affected by off-target probe binding based on sequence similarity, supported by spatial and single-cell comparative analyses using Xenium with serial section datasets from Visium CytAssist and 3′ single-cell RNA-seq, respectively. We further identified potential off-target probe binding affecting custom Xenium gene panels. To assist in the interpretation of existing probe-based gene expression data as well as future probe design, we provide OPT as a software tool for predicting potential off-target probe binding. For future reference, we have run OPT on all publicly available 10x Genomics pre-designed Xenium gene panels and supply them as a ZIP file (*Supplementary file 10*).

Although we have predicted off-target binding based on sequence alignment, its effect on gene expression quantification may still vary. One reason is that the off-target protein- or non-protein-coding gene may not be expressed (*Appendix 1—figure 11*). For example, in the Xenium human breast gene panel, although *ADH1B* probes have predicted off-target binding to *ADH1A* and *ADH1C* based on perfect sequence homology, the sparse expression of *ADH1A* and *ADH1C* in both the Visium and scRNA-seq breast cancer data led to only a minor difference in the aggregated expression and quantitative results (*Appendix 1—figure 12*). Our analysis of HuBMAP custom probe panels demonstrated how to evaluate for the potential impact of predicted off-targets by integrating tissue-specific single-cell or bulk RNA-seq data from reference atlases (*Appendix 1—figures 9 and 10*). Overall, we anticipate evaluating whether predicted off-target genes are expressed in a tissue-specific manner will be useful for gauging whether predicted off-target binding is likely to meaningfully affect observed gene expression and interpretation when applied to a tissue of interest.

Other sources of non-specific signal may also arise, including probe self-hybridization or probe-probe interactions (*Appendix 1—figure 6B*). In general, probe binding specificity is influenced by numerous factors, with many methods previously developed to aid in the design of probe target sequences while taking these factors into consideration (*Wang and Seed, 2003*; *Rouillard et al., 2003*; *Wernersson and Nielsen, 2005*; *Chou, 2010*; *Li et al., 2011*; *Hu et al., 2020*; *Hershberg et al., 2021*; *Fornace et al., 2022*; *Stenberg et al., 2005*; *Kuemmerle et al., 2024*). For example, in the Xenium human breast gene panel, *HDC*, a custom gene not included in the pre-designed Xenium v1 Human Breast Gene Expression Panel, did not have any off-targets predicted by OPT. Yet in Xenium, *HDC* exhibited a distinct spatial pattern and high global expression level, whereas in both Visium and scRNA-seq, *HDC* showed a minimal spatial pattern and sparse expression level, respectively (*Appendix 1—figure 13*). This illustrates how discrepancies across platforms can signal potential off-target activity not captured by alignment-based predictions alone and highlights the general importance of experimental validation with orthogonal technologies since sequence alignment-based tools such as OPT may not flag all potential discrepancies.

We note that most off-target binding impacts paralogs, homologous genes that have diverged following gene duplication events. Paralogs often belong to large gene families whose members can share high sequence similarity, increasing the risk of off-target probe activity. Unlike orthologs with conserved functions across species, paralogs are additional copies that can acquire function-altering

mutations (*Platt et al., 2000*; *Pevny et al., 1991*; *Tsai et al., 1994*). Pooling expression signals across paralogs can therefore prevent researchers from capturing their distinct functional roles.

We also found that 10 of the 2582 Xenium human breast gene panel probe target sequences did not align to any reference transcripts in the GENCODE v47 annotation. Upon manually aligning these probes to the GRCh38 genome, we determined that each unmapped sequence corresponded either to regions immediately upstream or downstream of annotated transcripts, or to intronic sequences that would not typically be present in mature RNA. Interestingly, when we aligned several of these probes, such as (ENSG00000125878|TCF15|5d3cbc2) and (ENSG00000169083|AR|a0c6719), to an earlier annotation (GENCODE v28), they instead mapped to exonic regions (*Appendix 1—figure 14*). This suggests that these intronic or intergenic probe target sequences were likely designed using older GENCODE versions in which those regions were annotated as exons. This finding illustrates the importance of disclosing the specific annotation version to promote reproducibility, as well as the ongoing variability of human gene annotation. Likewise, as evidenced by our analysis across GENCODE, RefSeq, and CHESS, we emphasize the variation across these reference annotations and therefore recommend using multiple annotations when designing probes and evaluating them for off-target effects to ensure a more comprehensive assessment.

Given these challenges, we advise probes with predicted off-target binding to protein-coding genes based on high sequence homology be avoided in future experiments. Likewise, we encourage the use of tools like OPT to aid in future probe design decisions and help ensure that probes are optimized to minimize off-target binding based on the most current transcriptome annotations. When probes with predicted off-targets cannot be avoided, we encourage the integration of tissue-specific RNA-seq data from HuBMAP and other reference atlases to evaluate for its potential impact. Further, such integration of tissue-specific RNA-seq data from reference atlases into the probe design process itself may offer a data-driven opportunity to minimize the impact of potential off-target binding by enforcing stricter probe-design constraints only where potential off-target genes are highly expressed in the tissue of interest. For datasets that have already been generated using probes with predicted off-target binding, we generally recommend taking into consideration these predictions to avoid drawing misleading conclusions. For example, we recommend expression measurements for genes with predicted off-target binding be omitted from training foundation models to avoid error propagation. Alternatively, when performing integrative analyses that compare or align gene expression with measurements across technologies, it may be necessary to incorporate off-target binding predictions. For instance, integration could be performed between the observed Xenium gene expression and the aggregated expression of the target and predicted off-target genes for the orthogonal technology. Finally, existing literature that base conclusions on genes with predicted off-target binding should be interpreted with caution.

We emphasize that these findings were missed in the previous Janesick et al. publication from 10x Genomics (*Janesick et al., 2023*). Consistent with previously published observations, we observed a highly correlated total gene expression magnitude between Xenium and Visium as well as scRNA-seq. However, a notable exception is *APOBEC3B*, which is not expressed according to both Visium and scRNA-seq but highly expressed according to Xenium (*Figures 2B and 3B*) – a discrepancy that Janesick et al. omitted. We emphasize that positive significant average gene expression correlation is a necessary but not sufficient metric for consistency across technologies and that individual data points should be scrutinized. Likewise, validation with orthogonal technologies could have helped identify discrepancies suggestive of off-target effects. We note Janesick et al. used immunofluorescence to validate two genes, *ERBB2* and *MS4A1*, which by our analysis were predicted to exhibit no off-target binding. Although Xenium incorporates blank and negative control probes that are intended to help quantify the rate of non-specific and potential off-target binding, our findings suggest that relying solely on such probes for error detection may be insufficient. Implementing probe redundancy, where the same gene is targeted using different codewords, could provide an additional internal control to enable the detection of off-target binding.

Although we focus here on the 10x Genomics Xenium technology, we do not exclude the possibility that off-target binding may similarly affect other probe-based gene detection approaches from other commercial vendors. Any technology that relies on hybridization-based detection is inherently susceptible to off-target probe binding when sequence similarity exists. Further, hybridization-based detection often inherently involves a trade-off between sensitivity and specificity. Given these inherent

technological limitations, we therefore emphasize the importance of transparency through sharing probe target sequences at minimum. However, many companies do not release the probe target sequences used in their assays, limiting the consumer's ability to fully interpret their results as well as the community's ability to effectively characterize and benchmark performance variation across platforms. Therefore, we strongly recommend that companies publish probe target sequences for pre-designed panels and likewise that researchers using these technologies should obtain and publish probe target sequences used in their studies to support transparent and reproducible science.

This is not the first instance in which a commercially available platform has encountered challenges in probe design (*McCartney et al., 2016*; *Harbig et al., 2005*; *Mecham et al., 2004*; *Liu et al., 2010*). These findings underscore the critical role of academic researchers toward ensuring the robustness of industry-led product development by providing oversight, free of financial conflicts of interest through independent federal funding. This complementarity between industry and academia fosters a more rigorous, transparent, and reliable scientific process, ultimately to the benefit of consumers and the public. By shedding light on putative off-target probe binding as well as by providing a tool to enable such off-target binding predictions, this work will help enhance the quality of spatial transcriptomics data and improve the overall reproducibility in spatial transcriptomics research.

## Methods
### OPT tool
OPT (Off-target Probe Tracker) is a Python program that runs nucmer (*Marçais and Kingsford, 2011*) for alignment and then processes the results to predict probe binding based on sequence homology. OPT is available as an open-source Python toolkit at https://github.com/JEFworks-Lab/off-target-probe-tracker, copy archived at *Hallinan et al., 2026*. When a user provides a query probe target sequence file, a target transcript sequence file, and the annotation used to extract these transcripts, OPT outputs which gene each probe is likely to bind to. Nucmer is a fast nucleotide sequence aligner that uses maximal exact matches as anchors, which it then joins together to find longer alignments. By default, OPT saves nucmer results in SAM format and finds perfect sequence matches between a query probe and a target transcript, requiring that alignments consist of only matches and cover the entirety of the query. OPT consists of four modules: (1) `flip` for reverse complementing probe target sequences aligned to the opposite strand of their target genes; (2) `track` for aligning probe target sequences and processing alignment results; (3) `stat` for compiling summary statistics on the number of off-target binding probes and affected genes; and (4) `all` for running the `flip`, `track`, and `stat` modules at once.

In the case that a probe's target gene has synonyms, we consider alignments to genes annotated with one of its synonyms to still be on-target. For example, if a probe that targets *NARS* shows alignments to a gene called *NARS1*, we don't consider it to be off-target binding. We gathered relevant gene synonym relationships using the GeneCards and HGNC online database.

OPT also provides a 'pad' mode in which imperfect alignments are allowed at either end of the query (i.e., probe target sequence). The -pl parameter sets the pad length at either end of the query, and OPT allows for any number of mismatches in these padded regions. For example, if the pad length is 10 and the probe target sequence length is 40 bp, then the middle 20 bp are the only part of the probe target sequence required to match. As long as the critical region is intact, OPT reports an off-target binding site based on this alignment. By default, -pl is set to 0, and the pad mode is activated by providing a non-zero integer to -pl.

### Obtaining probe target sequences for the Xenium v1 human breast gene expression panel
To identify potential off-target binding impacting the 10x Genomics Xenium v1 Human Breast Gene Expression Panel, we obtained the FASTA file of probe target sequences from the Janesick et al. publication courtesy of 10x Genomics and available as *Supplementary file 1* for preservation. Notably, this panel slightly deviates from the commercially available Xenium v1 Human Breast Gene Expression Panel (Appendix Note).

The target gene names and IDs were extracted from the probe IDs of the following format:
```
> gene_id|gene_name|accession
```

We expected the provided FASTA file to contain probe target sequences to be the reverse-complemented sequence of their intended target genes and hence align to the reverse strand of their target isoforms. However, when we aligned the breast panel probe target sequences to the GENCODE basic (v47) reference transcripts using nucmer, we found that 2563/2582 of probe target sequences aligned on the reverse strand of their target transcripts (i.e., isoforms of their target genes). For consistency, we enforced that all probe target sequences be oriented in the same direction and align to the forward strand of their target genes and transcripts. As such, we reverse-complemented these 2563 probe target sequences. We also added this functionality as an OPT module called 'flip' in which probe target sequences aligned to the reverse strand of their targets are reverse comple-mented. We expect probe target sequences to align to the forward strand of transcripts (i.e., both oriented in the same direction) during the downstream probe target sequence binding prediction step. The Xenium dataset, collected from a breast cancer tissue block utilized in Janesick et al., was down-loaded from the 10x Genomics website (https://www.10xgenomics.com/products/xenium-in-situ/preview-dataset-human-breast).

## Visium comparison

The Visium CytAssist dataset, collected from a breast cancer tissue block utilized in Janesick et al., was also downloaded from the 10x Genomics website (https://www.10xgenomics.com/products/xenium-in-situ/preview-dataset-human-breast). This dataset originally contained 4992 spots with $x$–$y$ coordinates and included 18,085 genes per spot. Of the 313 unique genes in the Xenium dataset, 307 were shared with the Visium dataset; the other six genes (*AKR1C1*, *ANGPT2*, *BTNL9*, *CD8B*, *POLR2J3*, and *TPSAB1*) were excluded from the analysis because they were absent from the Visium dataset.

To compare spatial gene expression patterns from Visium and Xenium technologies, we first mapped all the data to the same coordinate space. We used STalign (v1.0.1), a computational tool that utilizes affine transformations along with diffeomorphic metric mapping to align target and source datasets (*Clifton et al., 2023*). The initial alignment involved only affine transformations and eight manually determined landmarks to align the Visium histology image (source) to the Xenium histology image (target). This transformation brought the Visium image into the coordinate space of the higher-resolution Xenium image. We then applied this learned transformation to the Visium spots, ensuring that they were correctly positioned relative to both histology images. Next, we used STalign to map the Xenium transcripts (source) onto their corresponding Xenium histology image (target) using both affine and diffeomorphic metric mapping. The transcripts were rasterized at 30 µm resolution, with an initial affine transformation guided by four manually defined landmarks. Diffeomorphic metric mapping was then performed with the following parameters: a = 2500, epV = 1, niter = 2000, sigmaA = 0.11, sigmaB = 0.10, sigmaM = 0.15, sigmaP = 50, muA = [1, 1, 1], muB = [0, 0, 0], with all other settings left at their defaults. We extracted the overlapping regions between the two datasets (*Appendix 1—figure 3A*), which reduced the total spots in the Visium dataset to 3958. Finally, we aggregated the Xenium gene expression data to ~55 µm × 55 µm patches that correspond to the spatial locations of the Visium spots, resulting in matched-resolution spatial gene expression for both technologies (*Appendix 1—figure 3B*). The Visium spatial gene expression data is displayed as patches rather than spots to enhance visual saliency and ensure consistency with the Xenium spatial gene expression plots.

After obtaining matched-resolution spatial gene expression matrices, we quantified agreement between the Visium and Xenium data. For each gene shared between the two platforms, we constructed expression vectors across the aligned spatial spots, where each element corresponded to the log-normalized gene expression at its matched location in the tissue section. We then compared the Visium and Xenium vectors for each gene using two metrics: RMSE, computed relative to the line $y = x$, and Pearson correlation ($r$). To assess whether discrepancies in Xenium measurements could be attributed to predicted off-target genes, we compared the Xenium data of a target gene to the aggregated Visium expression of the target genes with its predicted off-targets. We summed the raw Visium counts of all off-target genes associated with each Xenium gene with predicted off-targets, re-normalized the aggregated counts using counts per million (CPM) followed by log($x$ + 1), and again calculated the RMSE and Pearson correlation. This approach enabled us to evaluate whether Xenium expression patterns aligned more closely with the intended target gene alone or with the combined expression of its predicted off-target genes.

## Single-cell RNA-seq comparison

The Chromium Next GEM 3′ scRNA-seq dataset, collected from a breast cancer tissue block utilized in Janesick et al., was downloaded from the 10x Genomics website (https://www.10xgenomics.com/products/xenium-in-situ/preview-dataset-human-breast). This dataset contained 12,388 cells with 36,601 genes per cell. All 313 unique genes present in the Xenium dataset are also in the scRNA-seq dataset; hence, both datasets were subsetted to these genes for the analysis.

Both scRNA-seq and Xenium provide single-cell resolution data. To integrate these datasets, we first removed cells lacking detectable gene expression. We then normalized the combined gene expression data using CPM and applied a log transformation with a pseudocount of 1. Principal component analysis is then applied to the normalized data, and batch effects are corrected using Harmony (v1.2.3) on the top 30 PCs using default parameters except for theta, which was set to 8, to promote further mixing with clusters across technologies. Finally, UMAP is performed on the harmonized PCs, generating a shared 2D embedding across the two technologies, and the data is further facetted by technology for visualization (*Appendix 1—figure 5*).

After generating a shared embedding, we quantified differences in gene expression patterns between scRNA-seq and Xenium. We first computed Leiden clusters on the harmonized PCs (resolution = 1.0) to identify transcriptionally similar groups of cells shared across both technologies. For each Leiden cluster, we calculated the mean expression of every gene present in both datasets. Clusters containing fewer than ten cells from either modality were excluded to ensure robust gene-level estimates. To compare expression patterns between scRNA-seq and Xenium for a given gene, we constructed vectors of cluster-level mean expression from both technologies and evaluated their similarity using two metrics: RMSE, computed relative to the line $y = x$, and Pearson correlation ($r$). To investigate whether predicted off-target genes may contribute to the observed target-gene expression in Xenium, we compared the Xenium target gene to the aggregated scRNA-seq expression of itself and its predicted off-targets. We summed the raw counts of a gene and all of its predicted off-targets in the scRNA-seq dataset, re-normalized the aggregated counts using CPM followed by log($x$ + 1), and again calculated the RMSE and Pearson correlation. This allowed us to test whether the Xenium gene expression more closely reflected the intended gene alone or the combined expression of the intended gene and its predicted off-targets.

## Obtaining custom probe panels from HuBMAP

To evaluate potential off-target binding in the HuBMAP placenta and multi-tissue (heart, kidney, and lung) custom probe panels, we first downloaded the corresponding BED files from the HuBMAP portal (https://portal.hubmapconsortium.org/browse/dataset/28fe8e4ac8a4193f82fdd9f4d4eb0bb2; https://portal.hubmapconsortium.org/browse/dataset/6f597ca43db80f2499443f5c5bfac97c). Using pyfaidx and pandas in Python, we extracted each probe's target gene name, gene identifier, and genomic coordinates from the BED files, and then generated FASTA files by retrieving the corresponding sequences from the reference genome (GRCh38). These FASTA files were then used as input to OPT to predict potential off-target binding.

## HuBMAP custom probe-panel evaluation

To assess whether predicted off-target genes were likely to confound Xenium results in the HuBMAP custom probe panels, we evaluated the expression of the predicted off-target genes in matched HuBMAP RNA-seq datasets corresponding to the tissues for which each panel was designed. Four RNA-seq datasets were downloaded from the HuBMAP portal: a bulk RNA-seq dataset for the placenta (https://doi.org/10.35079/HBM549.BBBQ.445) and scRNA-seq datasets for the heart (https://doi.org/10.35079/HBM378.WGXD.394), kidney (https://doi.org/10.35079/HBM793.TLPP.486), and lung (https://doi.org/10.35079/HBM826.BQLS.392).

For each tissue, raw count matrices were normalized to CPM and averaged across all cells to obtain a bulk-like mean expression profile. Log1p-transformed mean CPM values were used for all downstream comparisons. OPT was run on the two HuBMAP custom probe panels with all RNA species included and a pad length of 10. Since OPT reports transcript-level identifiers, we removed transcript-specific suffixes from Ensembl IDs to align them with gene symbols present in the RNA-seq datasets. For every target gene in the custom panels, we then compiled its predicted off-target genes based on OPT results and evaluated whether these off-targets were expressed in the matched tissue's RNA-seq

profile. To visualize these results, we generated heatmaps in which rows correspond to intended target genes and columns represent their predicted off-targets ordered by decreasing expression. This enabled direct comparison of the magnitude and tissue specificity of potential off-target expression across the HuBMAP datasets.

## Cross-annotation analysis

To compare OPT's results with different reference annotations, we used the most recent releases of GENCODE basic (v47), GENCODE comprehensive (v47), RefSeq (v110), and CHESS (v3.1.3) annotation of the GRCh38 genome. Note that GENCODE 'basic' is the more reliable version of the annotation and is much closer to RefSeq and CHESS. GENCODE 'comprehensive' includes hundreds of thousands of low-quality annotations, which we included in some of our analyses for completeness. Note also that GRCh38 has many non-reference sequences called 'alternative scaffolds'; we removed these for our analysis. We then used gffread to extract transcripts as defined in these annotations by running:

```
$ gffread -w transcripts.fa -g grch38.p12/14.fa annotation.gff
```

The GRCh38.p14 assembly was used during transcript sequence extraction for all reference annotations, except for CHESS which specifies that the annotation maps genes and transcripts onto the GRCh38.p12 assembly. For RefSeq, we renamed the VD(J) segment features as transcript features to ensure consistency, and we also removed transcript sequences with the gene_biotype 'pseudogene'. RefSeq has a separate biotype called 'transcribed_pseudogene', but does not annotate transcripts for these features. We considered transcripts annotated for a small subset of just pseudogenes an error in the annotation.

## Acknowledgements

We thank Reza Kalhor for his input on the project and feedback on the manuscript. We thank Ian Fiddes for sharing the Xenium gene panel from the Janesick et al. publication and explaining its relation to the commercially available pre-designed Xenium v1 Human Breast Gene Expression Panel. We thank Sergii Domanskyi, Scott Lindsay-Hewett, and Chenchen Zhu for sharing the HuBMAP custom gene panels. Research reported in this publication was supported by the National Institute of General Medical Sciences of the National Institutes of Health under Awards R35-GM142889 and R35-GM130151, the HuBMAP Integration, Visualization, and Engagement (HIVE) Initiative under Award Number OT2-OD033760, and the National Science Foundation under Grant No. 2047611.

## Additional information

### Funding

| Funder | Grant reference number | Author |
| --- | --- | --- |
| National Institute of General Medical Sciences | R35-GM142889 | Jean Fan |
| National Institute of General Medical Sciences | R35-GM130151 | Steven L Salzberg |
| National Institutes of Health | OT2-OD033760 | Jean Fan |
| U.S. National Science Foundation | 2047611 | Jean Fan |

The funders had no role in study design, data collection, and interpretation, or the decision to submit the work for publication.

### Author contributions

Caleb Hallinan, Conceptualization, Data curation, Software, Formal analysis, Validation, Investigation, Visualization, Methodology, Writing – original draft, Writing – review and editing; Hyun Joo Ji, Data curation, Software, Formal analysis, Investigation, Methodology, Writing – review and editing;

Edmund Tsou, Formal analysis, Validation, Visualization, Methodology, Writing – review and editing; Steven L Salzberg, Supervision, Methodology, Project administration, Writing – review and editing; Jean Fan, Conceptualization, Data curation, Supervision, Funding acquisition, Methodology, Writing – original draft, Project administration, Writing – review and editing

**Author ORCIDs**
Caleb Hallinan (iD) https://orcid.org/0009-0000-9137-1293
Hyun Joo Ji (iD) https://orcid.org/0009-0008-4360-5428
Edmund Tsou (iD) https://orcid.org/0009-0008-7339-465X
Steven L Salzberg (iD) https://orcid.org/0000-0002-8859-7432
Jean Fan (iD) https://orcid.org/0000-0002-0212-5451

Reviewer #2 (Public review): https://doi.org/10.7554/eLife.107070.3.sa1
Reviewer #3 (Public review): https://doi.org/10.7554/eLife.107070.3.sa2
Author response https://doi.org/10.7554/eLife.107070.3.sa3

---

## Additional files

### Supplementary files

Supplementary file 1. Fasta file of probe target sequences from 10x Genomics corresponding to the 10x Genomics Xenium v1 Human Breast Gene Expression Panel used in Janesick et al.

Supplementary file 2. Off-target Probe Tracker (OPT) output of genes with predicted off-target binding based on perfect sequence homology in RefSeq. This table shows the 14 genes whose probes in the 10x Genomics Xenium v1 Human Breast Gene Expression Panel exhibit predicted off-target probe binding, where each off-target alignment involves a perfect 40 bp match to the probe target sequence. The final column shows the gene types, in order, of each of the off-target genes shown in column 3. Off-target alignments between CCPG1 probes and DNAAF1-CCPG1 were excluded. Abbreviations: PC = protein-coding; PG = pseudogene; precursor_RNA = precursor RNA; misc_RNA = miscellaneous RNA; ncRNA = non-coding RNA.

Supplementary file 3. Off-target Probe Tracker (OPT) output of genes with predicted off-target binding based on perfect sequence homology in CHESS. This table shows the 23 genes whose probes in the 10x Genomics Xenium v1 Human Breast Gene Expression Panel exhibit predicted off-target probe binding, where each off-target alignment involves a perfect 40 bp match to the probe target sequence. The final column shows the gene types, in order, of each of the off-target genes shown in column 3. Off-target alignments between CCPG1 probes and DNAAF1-CCPG1 were excluded. Abbreviations: PC = protein-coding; PG = pseudogene; miRNA = microRNA.

Supplementary file 4. The number of off-target probes and affected genes (from the set of 313 genes in the Xenium panel) found when looking for perfect matches between probe target sequences and transcripts in four different reference annotations: GENCODE basic, GENCODE comprehensive, RefSeq, and CHESS. Off-target alignments between CCPG1 probes and DNAAF1-CCPG1 were excluded.

Supplementary file 5. Union set of protein-coding genes that Off-target Probe Tracker (OPT) predicts to be affected by off-target binding, across three different reference annotations: GENCODE basic, RefSeq, and CHESS.

Supplementary file 6. Eighteen additional genes that were identified to exhibit potential off-target probe binding when allowing for a 10-bp error margin on either side of the binding site when using GENCODE v47. Abbreviations: PC = protein-coding; PG = pseudogene; lncRNA = long non-coding RNA.

Supplementary file 7. Ten genes that were previously predicted to be affected by off-target binding based on perfect matching now show additional predicted off-target interactions when a 10-bp error margin is allowed on either side of the binding site when using GENCODE v47. This effect is observed either through the accumulation of new probes with predicted off-target binding or via the identification of additional predicted off-target genes per probe. Abbreviations: PC = protein-coding; PG = pseudogene; lncRNA = long non-coding RNA.

Supplementary file 8. Off-target Probe Tracker (OPT) output for 49 genes with predicted off-target binding in the HuBMAP placenta custom probe panel, generated using GENCODE v47 and allowing a 10-bp mismatch on either end of each probe. The final column lists the gene types, in order,

corresponding to the off-target genes shown in column 3. Abbreviations: PC = protein-coding; PG = pseudogene; precursor_RNA = precursor RNA; misc_RNA = miscellaneous RNA; ncRNA = non-coding RNA.

Supplementary file 9. Off-target Probe Tracker (OPT) output for 24 genes with predicted off-target binding in the HuBMAP multi custom probe panel, generated using GENCODE v47 and allowing a 10-bp mismatch on either end of each probe. The final column lists the gene types, in order, corresponding to the off-target genes shown in column 3. Abbreviations: PC = protein-coding; PG = pseudogene; precursor_RNA = precursor RNA; misc_RNA = miscellaneous RNA; ncRNA = non-coding RNA.

Supplementary file 10. Off-target Probe Tracker (OPT) results for all publicly available 10x Genomics probe sets. Results include all possible RNA species, and using a pad length (-pl) of 10.

MDAR checklist

## Data availability

The current manuscript is a computational study, so no data have been generated for this manuscript. The Off-target Probe Tracker computational tool can be found on GitHub: https://github.com/JEFworks-Lab/off-target-probe-tracker, copy archived at *Hallinan et al., 2026*.

The following previously published dataset was used:

| Author(s) | Year | Dataset title | Dataset URL | Database and Identifier |
|---|---|---|---|---|
| Janesick M, Shelansky R, Gottscho A, Wagner F, Williams SR, Rouault M, Beliakoff G, Morrison CA | 2023 | High resolution mapping of the breast cancer tumor microenvironment using integrated single cell, spatial and in situ analysis of FFPE tissue | https://www.10xgenomics.com/products/xenium-in-situ/preview-dataset-human-breast | 10x Genomics, human-breast |

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

## Appendix 1

## Differences between three Xenium breast gene panels and the impact on interpretation of results

In the previous iteration of this manuscript, we used the publicly available Xenium v1 Human Breast Gene Expression Panel probe target sequences from the 10x Genomics website (prior to April 10, 2025) and assumed it was used to generate the Xenium breast cancer data from Janesick et al. based on the paper's methodological descriptions. 10x Genomics has since noted on their public website as well as through private communications that this publicly available file erroneously included extra probe target sequences that are not used. Further, the Xenium breast cancer data presented in Janesick et al. uses a human breast gene panel with probe target sequences that represent an earlier iteration of the commercially available Xenium v1 Human Breast Gene Expression Panel. In particular, compared to the commercially available panel, the Janesick et al. panel has 24 genes with one or more probes modified in the final panel, along with the probe target sequences targeting the 33 custom genes added by Janesick et al. This impacts our interpretation of observed results to provide stronger evidence of putative imperfect sequence homology based off-target probe binding.

For clarity, we refer to the three probesets as:

- Probeset A: the previously analyzed publicly available Xenium v1 Human Breast Gene Expression Panel probe target sequences (prior to April 10, 2025).
- Probeset B: the Xenium probe target sequences used in the Janesick et al. panel.
- Probeset C: the currently commercial pre-designed Xenium v1 Human Breast Gene Expression Panel (after April 10, 2025).

To provide a specific example, in the previous iteration of this manuscript, OPT identified a probe for *TUBB2B* with a target sequence that exhibited perfect sequence homology with *TUBB2A*:

>ENSG00000137285|TUBB2B|1dec8c0
GTTCATGATGCGGTCTGGGTACTCTTCCCGGATCTTGCTG

Analysis with orthogonal spatial and single-cell transcriptome profiling technologies suggested the *TUBB2B* gene expression pattern observed in the Xenium breast cancer data from Janesick et al. represented an aggregation of both *TUBB2B* and *TUBB2A* consistent with off-target binding. Given our assumption at the time that the Xenium breast cancer data had been generated using Probeset A, we therefore believed this perfect homology probe was responsible for the observed off-target signal.

However, we now understand that the Xenium breast cancer data from Janesick et al. were generated using Probeset B. Probeset B no longer contains this specific probe for *TUBB2B* with perfect sequence homology with *TUBB2A*. However, the observed *TUBB2B* gene expression pattern still represents an aggregation of both *TUBB2B* and *TUBB2A* based on analysis with orthogonal spatial and single-cell transcriptome profiling technologies. Although Probeset B no longer contains this specific probe for *TUBB2B* with perfect sequence homology with *TUBB2A*, it contains one probe with imperfect sequence homology with *TUBB2A* (=30 × 1 = 6 × 3):

>ENSG00000137285|TUBB2B|ed52e1c
TTGTCAATGCAGTAGGTTTCATCTGTGTTTTCCACCAGCT

The observed comparative gene expression patterns therefore provide newfound strong support for putative imperfect sequence homology off-target binding.

While we do not have Xenium breast cancer data generated using Probeset C, this *TUBB2B* probe target sequence is also present in Probeset C:

>TUBB2B (ENSG00000137285) | 3 | 5
TTGTCAATGCAGTAGGTTTCATCTGTGTTTTCCACCAGCT

This suggests that off-target effects may still impact commercially available pre-designed panels such as Probeset C (*Supplementary file 10*).

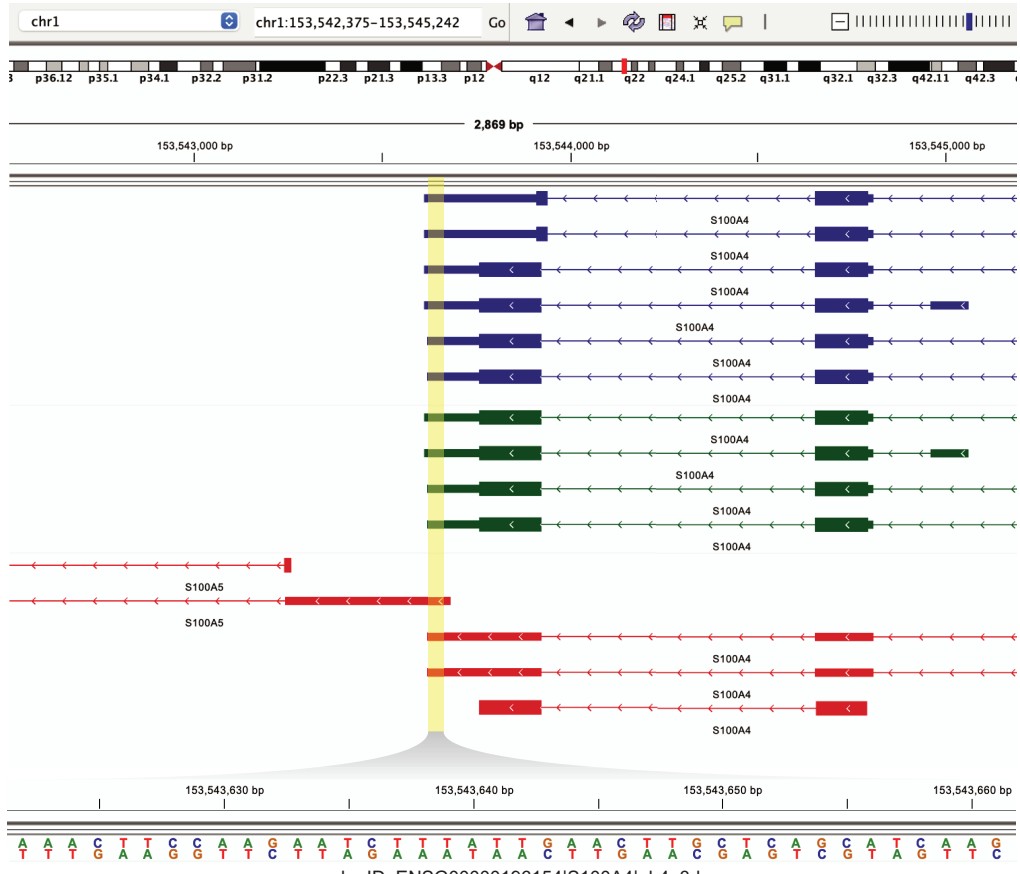

**Red: RefSeq    Blue: CHESS    Green: GENCODE**

**Appendix 1—figure 1.** Screenshot from the Integrated Genome Viewer (IGV) showing the 40 bp probe target sequence (ID: ENSG00000196154|S100A4|ab4e3dc) that matches both *S100A5* and *S100A4*. Shown are six isoforms from the CHESS v3.1 annotation, four from GENCODE basic v47, and three from RefSeq v110 for *S100A4*, as well as two RefSeq isoforms for the neighboring *S100A5*. The probe target sequence aligns to the overlapping region between *S100A5* and *S100A4* gene loci. Matching probe shown in a zoomed-in view below. The forward- and reverse-strand sequences of the probe are shown, and the highlighted area indicates approximately where the probe falls within the gene.

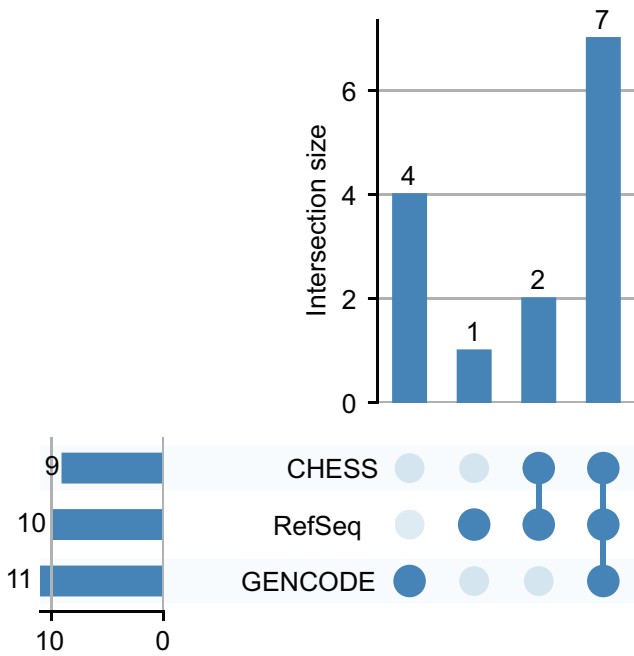

**Appendix 1—figure 2.** UpSet plot illustrating the overlap of protein-coding genes across three genome annotations: GENCODE basic, RefSeq, and CHESS.

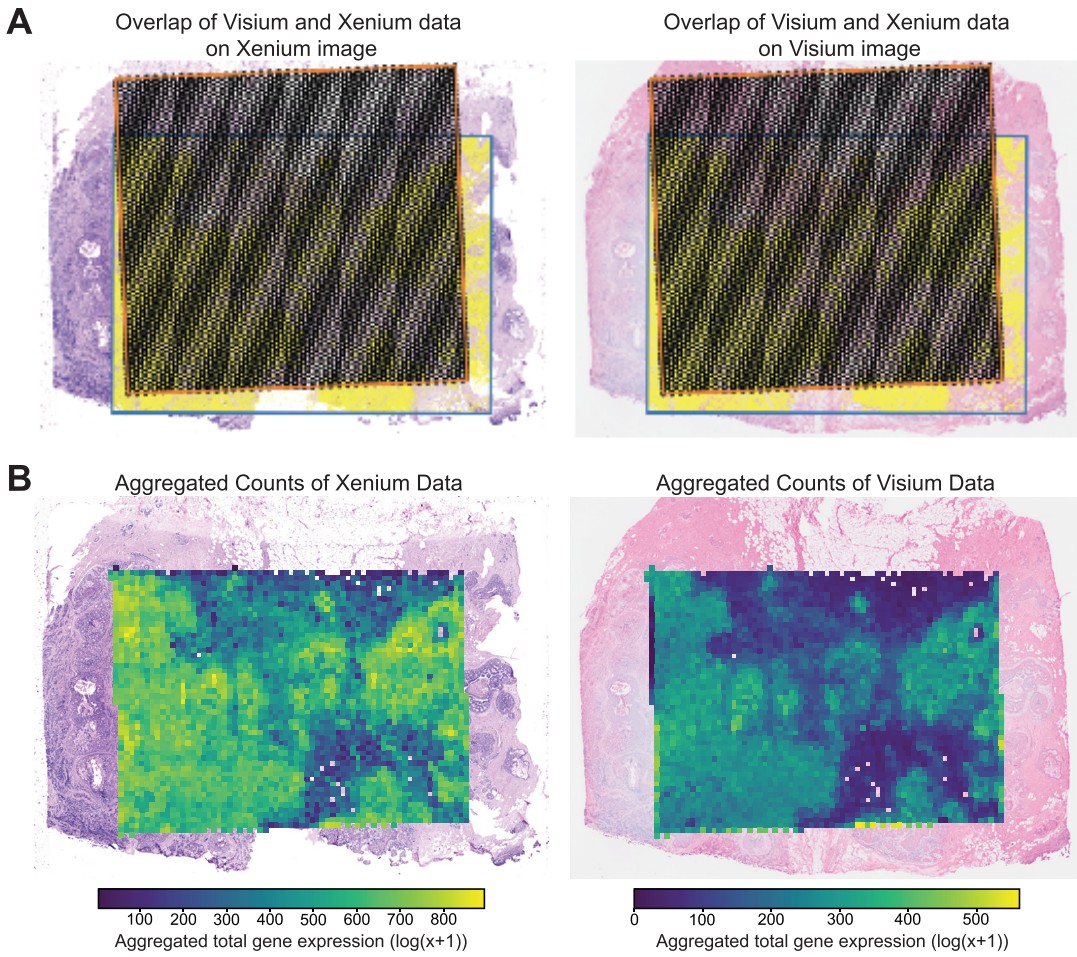

**Appendix 1—figure 3.** Preprocessing the Visium and Xenium datasets. (**A**) Overlap regions between Visium (orange outline) and Xenium (blue outline) data, shown on the Xenium histological image and the Visium histological image, respectively. (**B**) Log transformed aggregated total gene counts for spots (~55 µm × 55 µm) in both Xenium and Visium datasets, overlaid on their corresponding histological image.

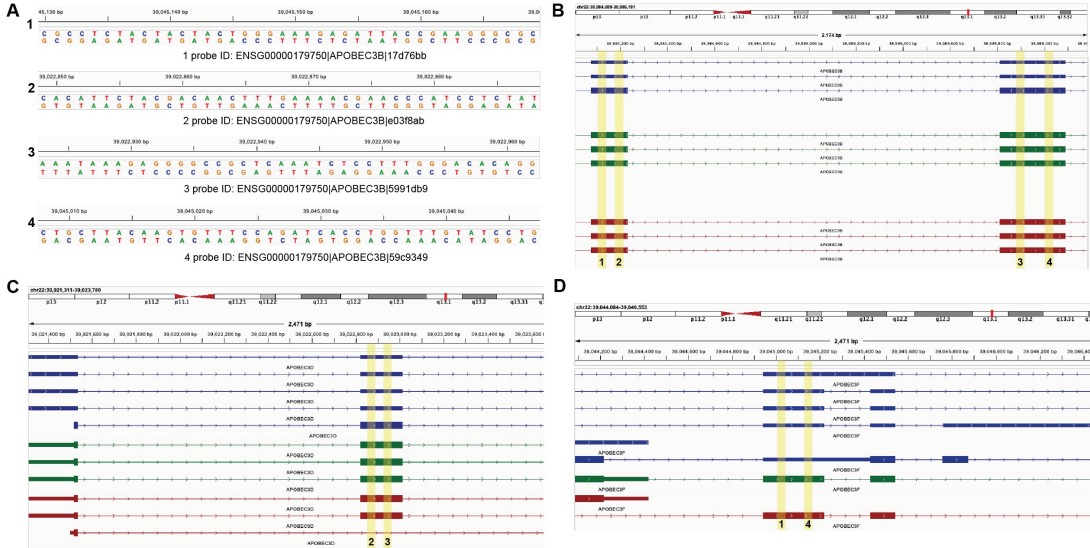

**Red: RefSeq   Blue: CHESS   Green: GENCODE**

**Appendix 1—figure 4.** Screenshot from the Integrated Genome Viewer (IGV) showing the 40 bp probe sequences that matches both APOBEC3B as well as APOBEC3D and APOBEC3F. (**A**) Four 40 bp probes targeting *APOBEC3B* (ENSG00000179750|APOBEC3B|17d76bb, ENSG00000179750|APOBEC3B|e03f8ab, ENSG00000179750|APOBEC3B|5991db9, and ENSG00000179750|APOBEC3B|59c9349). (**B**) All four probe target sequences align to their intended target gene *APOBEC3B*, while two of the four probe target sequences align to each off-target gene: (**C**) *APOBEC3D* and (**D**) *APOBEC3F*. The forward- and reverse-strand sequences of the probe target sequences are shown, and the highlighted areas indicate approximately where the probe target sequence falls within the gene. Panels (**B–D**) share a common legend.

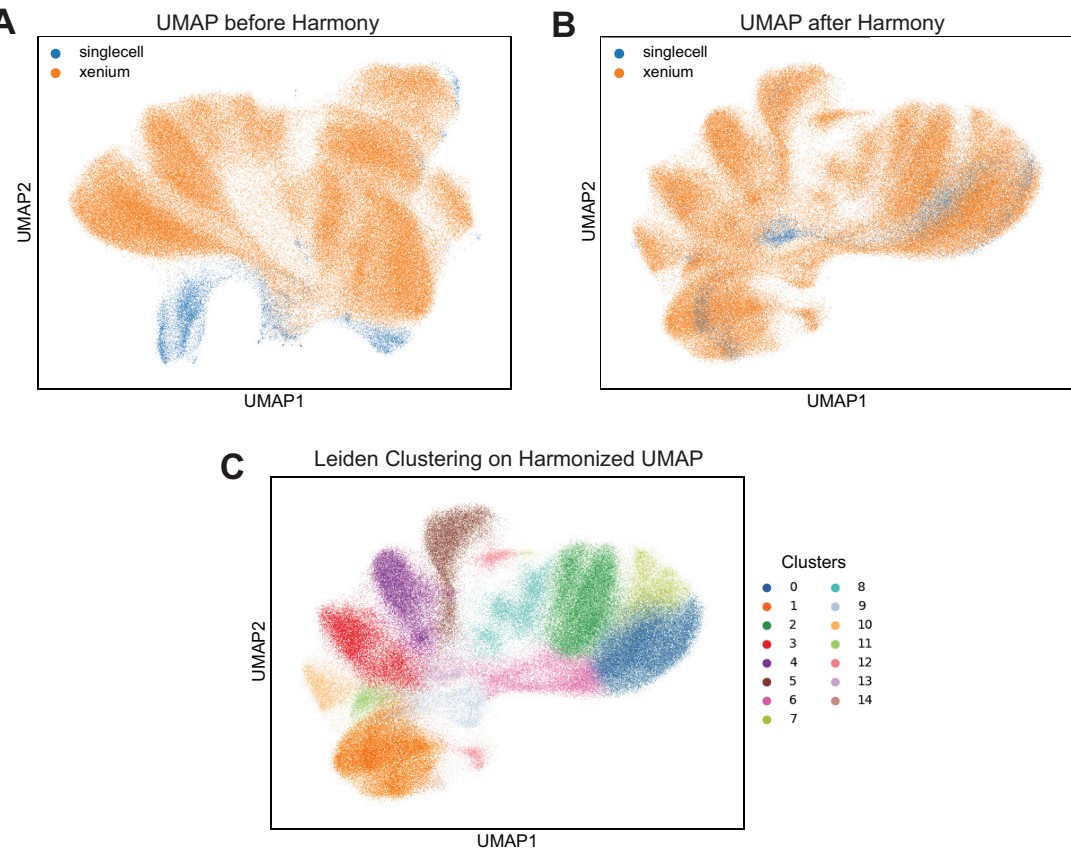

**Appendix 1—figure 5.** Uniform Manifold Approximation and Projection (UMAP) visualization of integrated single-cell RNA sequencing (scRNA-seq) and Xenium datasets. (**A**) Before harmony batch correction and (**B**) after harmony batch correction. (**C**) Leiden clustering results on the harmonized UMAP.

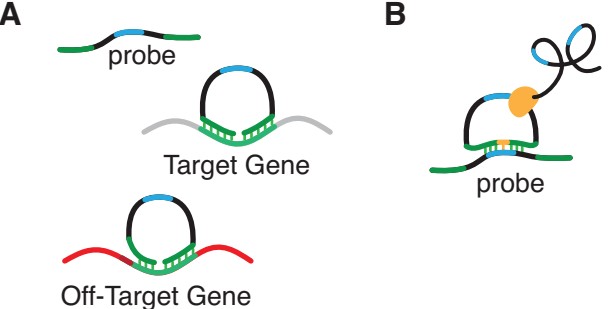

**Appendix 1—figure 6.** Illustrative schematics of potential probe binding issues. (**A**) Schematic illustrating that hybridization may still occur even when there is a sequence mismatch at the non-ligated ends of the probe target sequence. (**B**) Schematic depicting how probes could bind to each other instead of to their intended target.

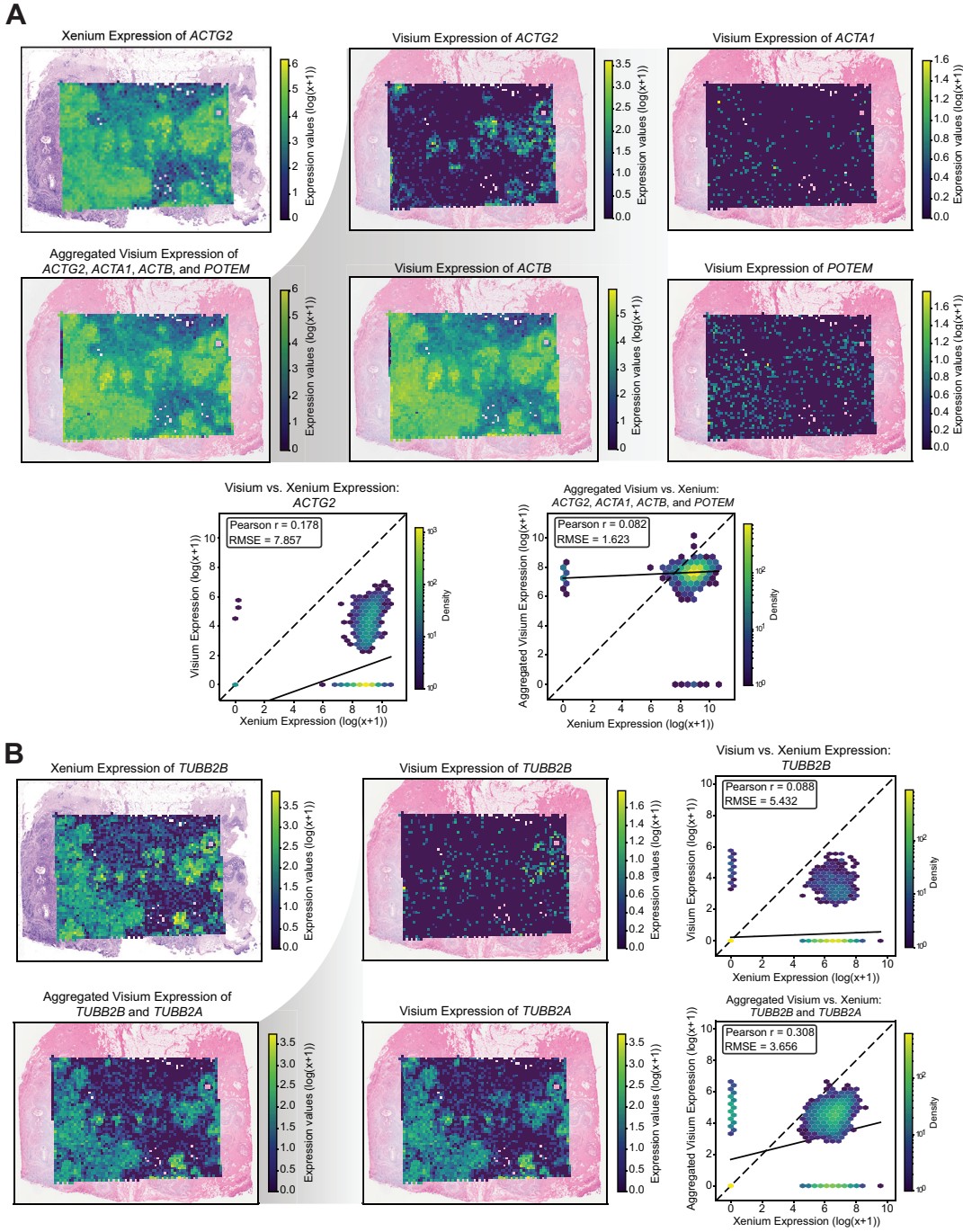

**Appendix 1—figure 7.** Effect of Predicted Off-Target Probe Binding on *ACTG2* and *TUBB2B* Expression Patterns using Visium data. (**A**) Gene expression patterns for *ACTG2*: Xenium expression, Visium expression, the aggregated Visium expression combining *ACTG2* and its predicted off-target gene's expression *ACTA1*, *ACTB*, and *POTEM*, and Visium expression of *ACTG2*'s predicted off-targets *ACTA1*, *ACTB*, and *POTEM*. Two density plots are shown: one comparing Xenium vs. Visium for *ACTG2* alone, and one comparing Xenium vs. the aggregated Visium expression. The dotted line indicates the identity line (*X* = *Y*), and the solid line represents the line of best fit. (**B**) Gene expression patterns for *TUBB2B*: Xenium expression, Visium expression, the aggregated Visium expression combining *TUBB2B* and its predicted off-target gene's expression *TUBB2B* and *TUBB2A*, and Visium expression of *TUBB2B*'s predicted off-target *TUBB2A*. Two density plots are shown: one comparing Xenium vs. Visium for *TUBB2B* alone, and one comparing Xenium vs. the aggregated Visium expression. The dotted line indicates the identity line (*X* = *Y*), and the solid line represents the line of best fit.

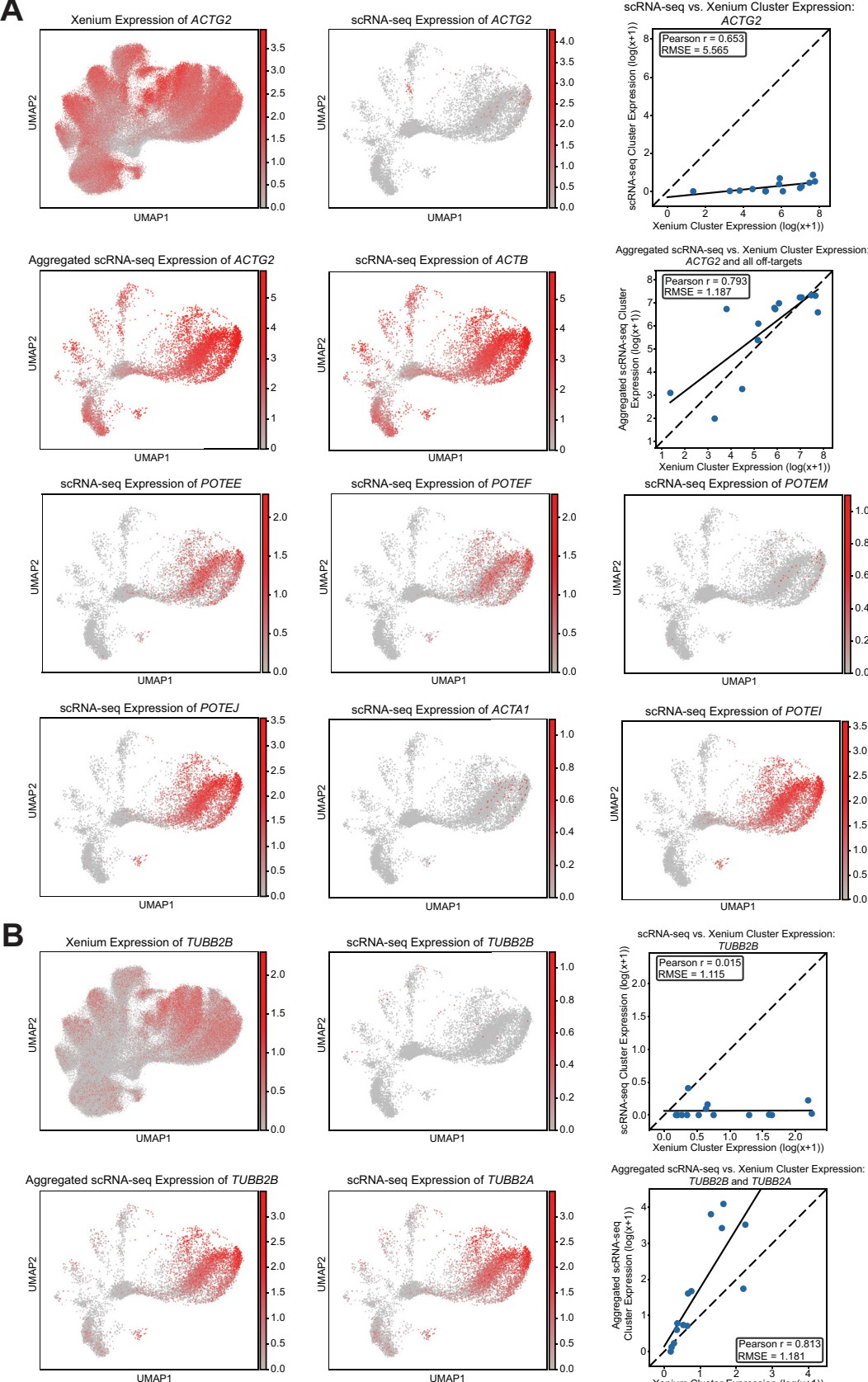

**Appendix 1—figure 8.** Effect of Predicted Off-Target Probe Binding on *ACTG2* and *TUBB2B* Expression

*Appendix 1—figure 8 continued on next page*

*Appendix 1—figure 8 continued*

Patterns using scRNA-seq data. (**A**) Comparison of *ACTG2* expression patterns on harmonized Uniform Manifold Approximation and Projection (UMAP): Xenium expression, single-cell RNA sequencing (scRNA-seq) expression, an aggregated scRNA-seq profile combining *ACTG2* and its predicted off-target gene's expression *ACTB*, *POTEM*, *POTEE*, *POTEF*, *POTEI*, *POTEJ*, and *ACTA1*, and scRNA-seq expression of *ACTG2*'s potential off-targets. Two scatterplots are shown: one comparing Xenium vs. scRNA-seq for *ACTG2* cluster expression alone, and one comparing Xenium vs. the aggregated scRNA-seq cluster expression. The dotted line indicates the identity line (*X = Y*), and the solid line represents the line of best fit. (**B**) Comparison of *TUBB2B* expression patterns on harmonized UMAP: Xenium expression, scRNA-seq expression, an aggregated scRNA-seq profile combining *TUBB2B* and its predicted off-target gene's expression *TUBB2A*, and scRNA-seq expression of *TUBB2B*'s potential off-target *TUBB2A*. Two scatterplots are shown: one comparing Xenium vs. scRNA-seq for *TUBB2B* cluster expression alone, and one comparing Xenium vs. the aggregated scRNA-seq cluster expression. The dotted line indicates the identity line (*X = Y*), and the solid line represents the line of best fit.

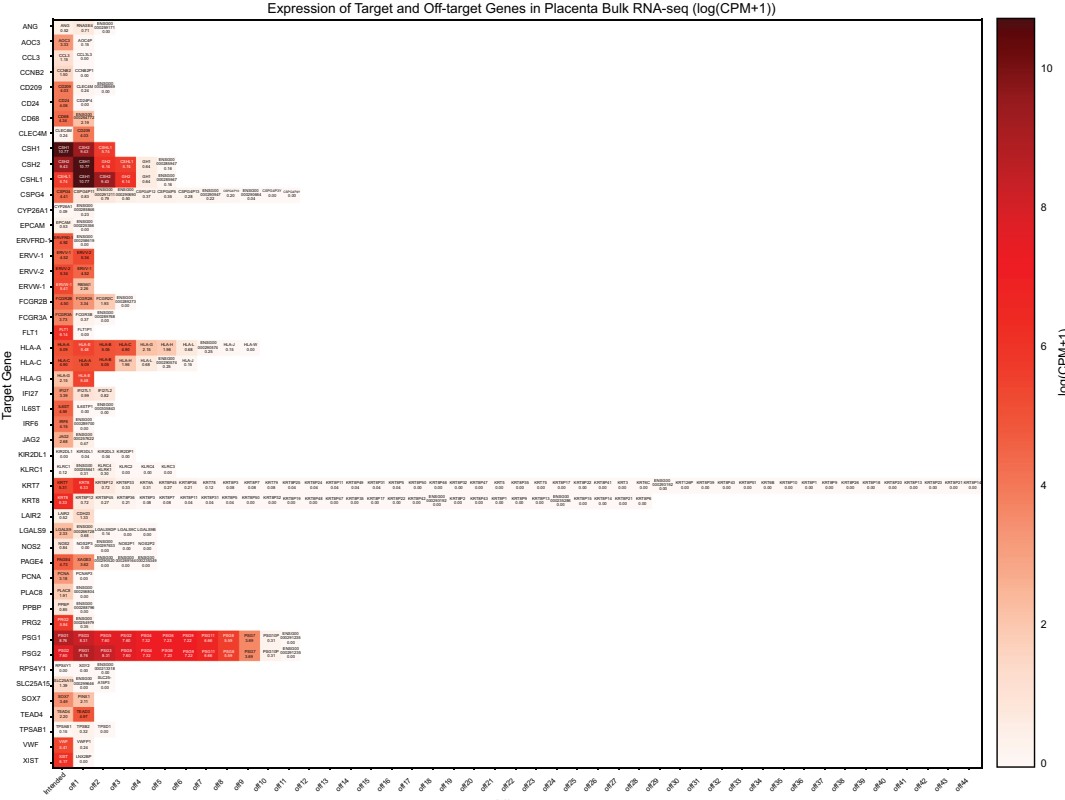

**Appendix 1—figure 9.** Heatmap visualization of target genes and their predicted off-target genes of the HuBMAP placenta custom probe panel using a corresponding placenta bulk RNA-seq dataset. Gene expression values are counts per million (CPM) normalized, with a pseudocount added prior to log transformation. The first column shows the expression of each target gene in the placenta bulk RNA-seq dataset, while the remaining columns display the expression of the corresponding predicted off-target genes.

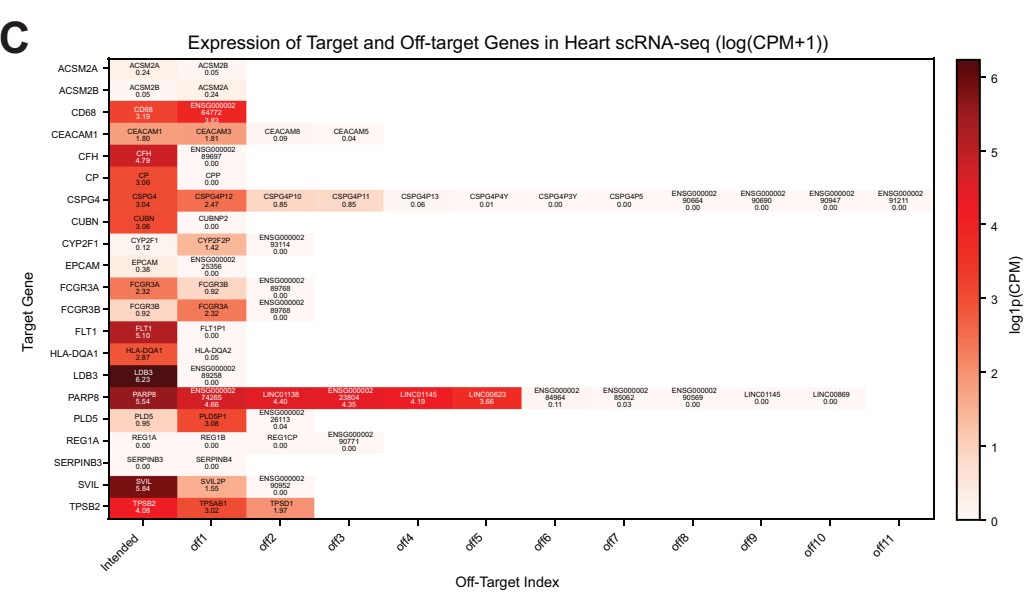

**Appendix 1—figure 10.** Heatmap visualizations of target genes and their predicted off-target genes of the HuBMAP multi custom probe panel using a corresponding (**A**) kidney, (**B**) lung, and (**C**) heart single-cell RNA sequencing (scRNA-seq) datasets. Gene expression values are counts per million (CPM) normalized, with a pseudocount added prior to log transformation. The first column shows the expression of each target gene in their respective scRNA-seq dataset, while the remaining columns display the expression of the corresponding predicted off-target genes.

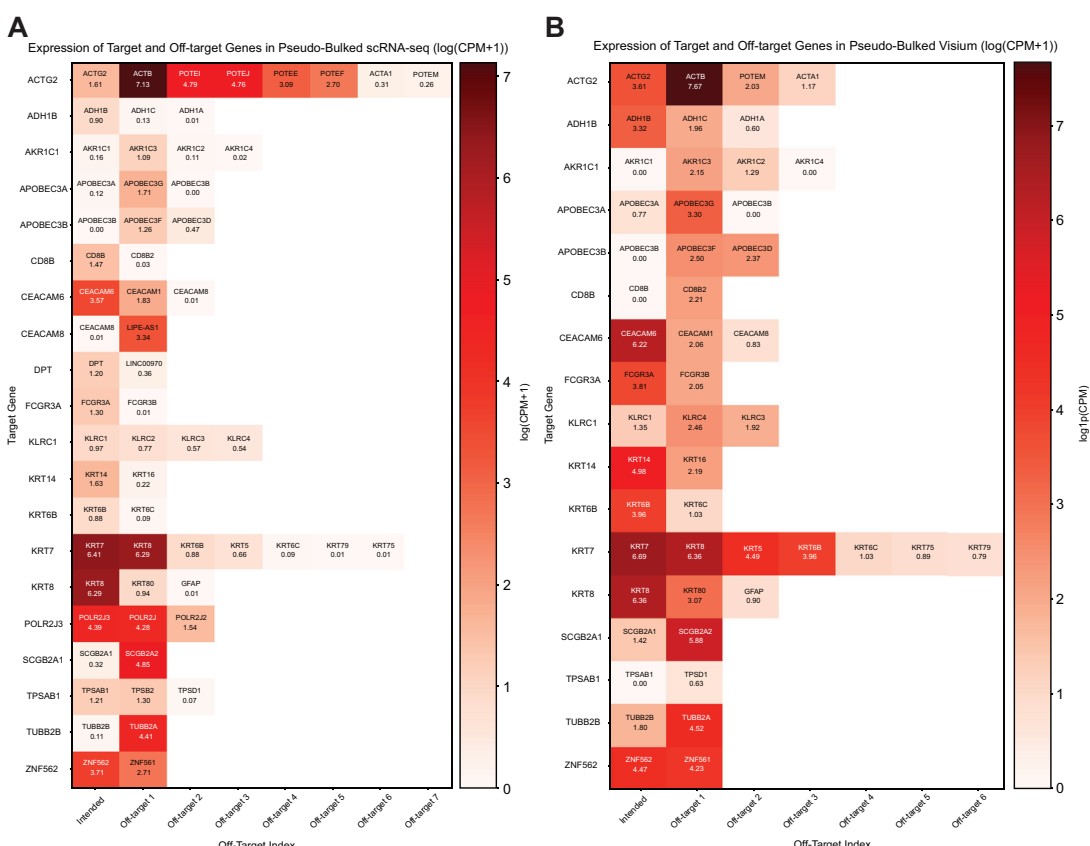

**Appendix 1—figure 11.** Heatmap visualizations of target genes and their predicted off-target genes for the Janesick et al. probes using (**A**) pseudo-bulked single-cell RNA sequencing (scRNA-seq) and (**B**) Visium data. Gene expression values are counts per million (CPM) normalized, with a pseudocount added prior to log transformation. The first column shows the expression of each target gene in their respective scRNA-seq dataset, while the remaining columns display the expression of the corresponding predicted off-target genes.

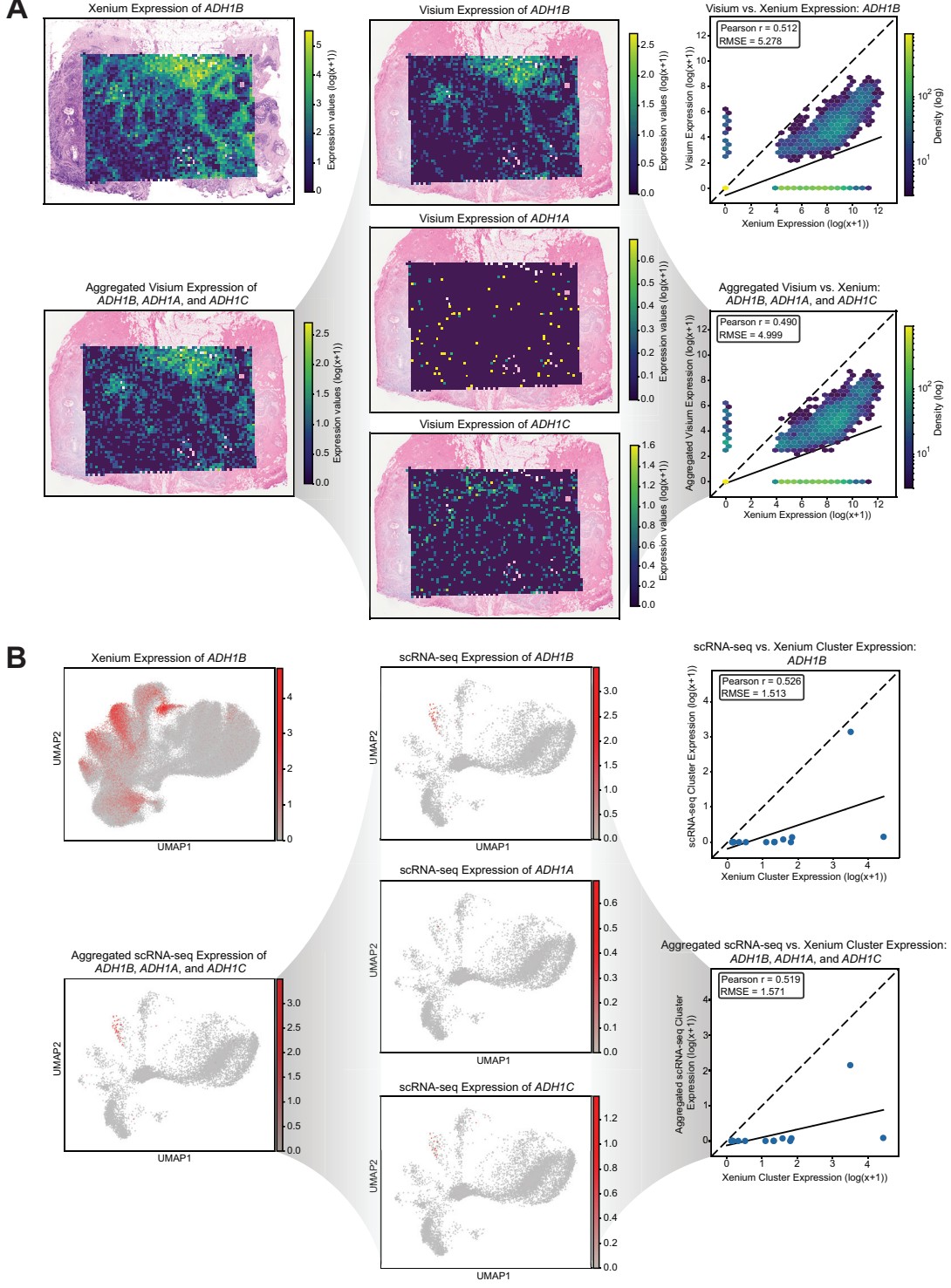

**Appendix 1—figure 12.** Effect of Predicted Off-Target Probe Binding on *ADH1B* Expression Patterns using Visium and scRNA-seq data. (**A**) Gene expression patterns for *ADH1B*: Xenium expression, Visium expression, the aggregated Visium expression combining *ADH1B* and its predicted off-target gene's expression *ADH1A* and *ADH1C*, and Visium expression of *ADH1B*'s predicted off-targets *ADH1A* and *ADH1C*. Two density plots are shown: one comparing Xenium vs. Visium for *ADH1B* alone, and one comparing Xenium vs. the aggregated Visium expression. The dotted line indicates the identity line (*X = Y*), and the solid line represents the line of best fit. (**B**) Comparison of *ADH1B* expression patterns on harmonized Uniform Manifold Approximation and Projection (UMAP): Xenium expression, single-cell RNA sequencing (scRNA-seq) expression, an aggregated scRNA-seq

*Appendix 1—figure 12 continued on next page*

*Appendix 1—figure 12 continued*

profile combining *ADH1B* and its predicted off-target gene's expression *ADH1A* and *ADH1C*, and scRNA-seq expression of *ADH1B*'s potential off-targets *ADH1A* and *ADH1C*. Two scatterplots are shown: one comparing Xenium vs. scRNA-seq for *ADH1B* cluster expression alone, and one comparing Xenium vs. the aggregated scRNA-seq cluster expression. The dotted line indicates the identity line ($X = Y$), and the solid line represents the line of best fit.

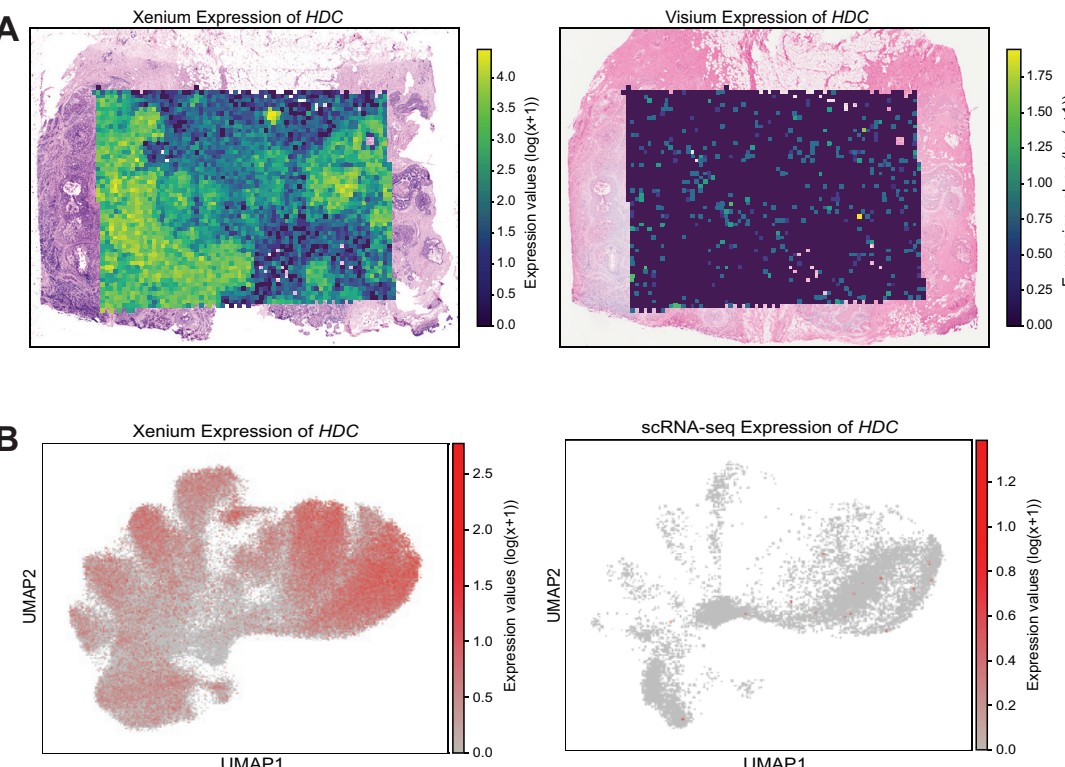

**Appendix 1—figure 13.** Spatial gene expression of *HDC* in Xenium, Visium, and scRNA-seq data. (**A**) Spatial gene expression of *HDC* overlaid on the corresponding histological images for Xenium and Visium. (**B**) Harmonized Uniform Manifold Approximation and Projection (UMAP) visualization of *HDC* expression for Xenium and single-cell RNA sequencing (scRNA-seq) data.

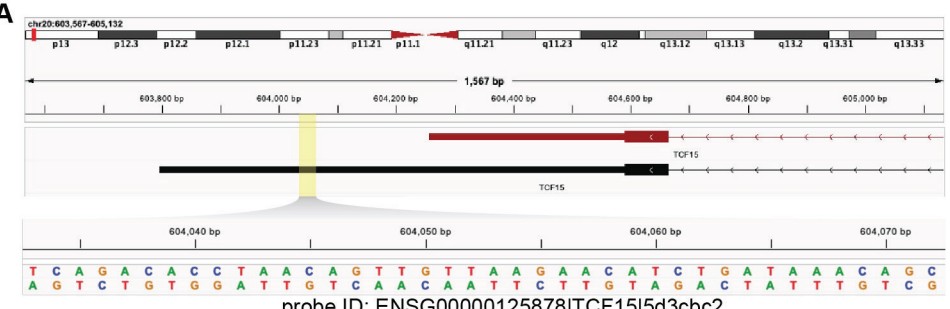

probe ID: ENSG00000125878|TCF15|5d3cbc2

**Red: GENCODE v47   Black: GENCODE v28**

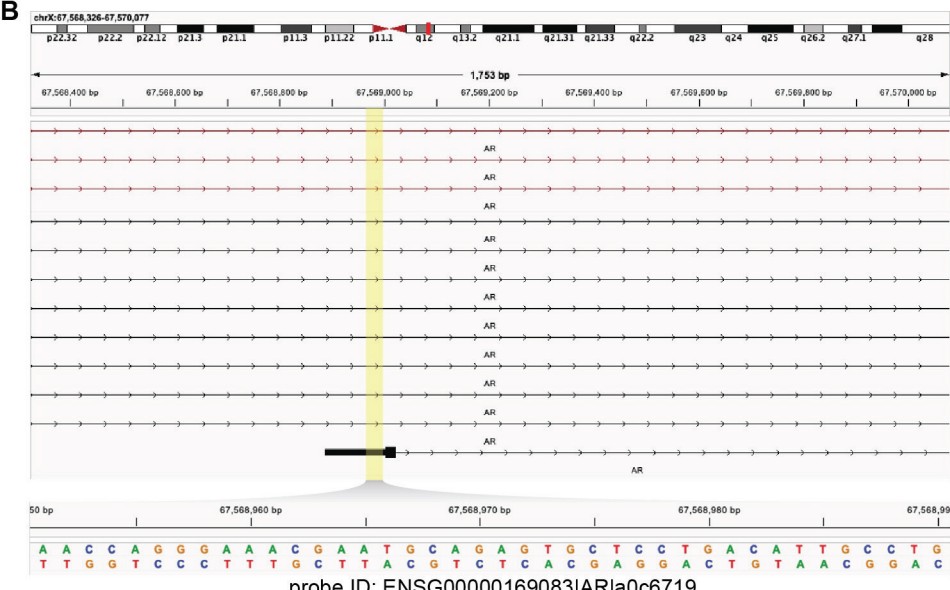

probe ID: ENSG00000169083|AR|a0c6719

**Appendix 1—figure 14.** Screenshots from the Integrated Genome Viewer (IGV) illustrating annotation-dependent differences in probe alignment. (**A**) A 40-bp probe (ID: ENSG00000125878|TCF15|5d3cbc2) aligns to an exonic region in GENCODE v28 but to an upstream region in GENCODE v47. (**B**) A 40-bp probe (ID: ENSG00000169083|AR|a0c6719) aligns to an exonic region in GENCODE v28 but to an intronic region in GENCODE v47. Matching probe shown in a zoomed-in view below. The forward- and reverse-strand sequences of the probe are shown, and the highlighted areas indicate approximately where the probe falls within the gene. Panels (**A, B**) share a common legend.

