## [Editor Report · eLife Assessment]

This **valuable** study identifies and characterizes probe binding errors in a widely used commercial platform for spatial transcriptomics, discovering that at least 14 out of 313 genes in a human breast cancer panel are not accurately detected. The authors provide **convincing** evidence for their findings through validation against multiple independent sequencing technologies and reference datasets, and they introduce a computational tool to help predict potential off-target probe binding. Given the broad adoption of this platform in biomedical research, this work provides an essential quality control resource that will improve data interpretation across numerous studies.

---

## [Referee Report · Reviewer #2 (Public review)]

This paper describes an analysis of a commercially available panel for a spatial transcriptomic approach and introduces a computational tool to predict potential off-target binding sites for the type of probe used in the aforementioned panel. The performance of the prediction tool was validated by examining a dataset that profiled the same cancer tissue with multiple modalities. Finally, a detailed analysis of the potential pitfalls in a published study communicated by the company that commercialized the spatial transcriptomic platform in question is provided, along with best practice guidelines for future studies to follow.

Strengths:

- The manuscript is clearly written and easy to follow.

- The authors provide clean, organized, and well-documented code in the associated GitHub repository.

Comments on revision:

My impressions from the first round of review haven't really changed. I don't think the software tool is well developed, and failing to incorporate thermodynamics or consider the impact of alignment settings is a major weakness.

I do think the topical area is relevant. The inclusion of the Xenium /Hubmap data modestly strengthens the manuscript relative to the original submission.

---

## [Referee Report · Reviewer #3 (Public review)]

Summary:

The authors present a new computational method (OPT) for predicting off-target probe binding in the commercial 10X Xenium spatial transcriptomics platform. They identified 28 genes in the 10x xenium human breast cancer gene panel (280 genes) that are not accurately detected at the single-molecule level. They validated the predicted off-target binding using reference data from single-cell RNA-seq and 3'-sequencing-based Visium RNA-seq. This work provides a practical resource and will serve as a valuable reference for future data interpretation.

Strengths:

(1) Provides a toolbox for the community to identify off-target probes.

(2) Validates the predictions using single-cell RNA-seq and sequencing-based Visium RNA-seq datasets.

Comments on revision:

The authors state that OPT is a new software tool and have posted example code on GitHub. However, the Jupyter notebook does not display any figures or workflows that would allow the process to be replicated. Please provide documentation and code that can reproduce the results/figures presented in the paper.

---

## [Author Response]

The following is the authors’ response to the original reviews

**Public Reviews:**

We thank the editors and the reviewers for their constructive feedback in helping us strengthen this manuscript.

During the revision process, new information was shared with us by the 10x Genomics team regarding the Xenium probe sequences evaluated in our original paper. Briefly, the Xenium probe sequences we evaluated represented an earlier iteration of the probes used to generate the data in Janesick et al. Further, we were made aware that the probe sequences used in Janesick et al. represented an earlier iteration of the commercially available Xenium v1 Human Breast Gene Expression Panel. We now elaborate further in a new Supplementary Note. We have therefore updated the paper throughout to reflect this new understanding, though we emphasize that our conclusions do not change. Rather, this newfound understanding provides stronger evidence of off-target probe binding with imperfect sequence matching, which we support with new supplementary figures.

(1) Limited evaluation of tissues and gene panels“The results were only tested with one tissue (human breast). However, this is not a major weakness, as one can easily extrapolate that this should be the case for any other tissue.”“Does not apply the OPT method to the most widely used Xenium gene panels (e.g., pan-Human, pan-Mouse panels with ~5,000 genes each).”“The authors claim that OPT is a generalizable method for identifying off-target probes. To support this claim, they should provide similar predictions for the Xenium Pan-Human or Pan-Mouse gene panels, which are more widely used than the breast cancer panel.”“While I understand that conducting new experimental studies is likely beyond the authors' intended scope of the manuscript, the narrow reliance on Janesick et al. for all of the validation makes it difficult to assess the broad usability of OPT. In the absence of designing and then validating novel padlock probe designs with OPT, are there other publicly available datasets that authors could perform secondary analysis on using OPT?”

Our primary focus on breast cancer was driven by data availability rather than tissue specificity. For this probe panel, matched Xenium, Visium, and scRNA-seq datasets are publicly available, enabling direct cross-platform comparisons of gene expression and allowing us to evaluate the impact of off-target probe binding in Xenium.

OPT is tissue-agnostic and can be applied to any probe panel regardless of tissue type. To demonstrate this generalizability, we have now applied OPT on all publicly available 10x Genomics probe sets beyond the breast panel, including the Xenium pan-Human and pan-Mouse gene panels. The complete results of these analyses have been generated and are provided as a compressed zip file accompanying the revised manuscript.

Beyond pre-designed panels, in this revision, we have now also applied OPT to custom Xenium gene panels from the Human BioMolecular Atlas Program (HUBMAP) and further demonstrate integration of HUBMAP RNA-seq data to evaluate the impact of potential predicted off-targets in a new section “Bulk RNA-seq reference atlases suggest off-target binding can variably impact results in Xenium custom probe panels.”

Overall, in these newly evaluated panels, we identify many cases of off-target probe binding with non-negligible expression of off-target genes in the target tissue, underscoring that our findings are not specific to human breast tissue. Therefore, in the revision, we have broadened the title to “Evidence of off-target probe binding affecting 10x Genomics Xenium Gene Panels compromise accuracy of spatial transcriptomic profiling”

(2) Limited quantifications“Lacks clarity on how the confidence level of off-target predictions is calculated.”“How can the confidence level of these off-target predictions be quantitatively assessed? Please provide benchmarks or validation metrics if available.”

We thank the reviewer for raising this important point. To strengthen our claim that predicted off-targets can contribute to observed Xenium expression patterns, we incorporated a quantitative assessment in addition to the qualitative comparisons presented previously. Specifically, we leveraged Visium and scRNA-seq data to compare spot- and cluster-level expression of target genes alone versus expression aggregated with their predicted off-target genes. Across all examples shown, inclusion of predicted off-targets consistently resulted in stronger agreement with the Xenium results, as reflected by decreased RMSE and increased Pearson correlation relative to using the target gene alone.

We emphasize, however, that OPT does not assign a formal confidence score to off-target predictions based on sequencing data alone. Importantly, identification of a potential off-target by OPT does not imply that it will necessarily affect Xenium results. As we’ve noted, if the off-target gene is not expressed, then it will not affect the observed gene expression magnitudes of the target gene. To help users assess whether predicted off-target genes will affect observed gene expression magnitudes of the target gene for a tissue of interest, we now provide a complementary analysis, including heat-map visualizations comparing the expression of target genes and their predicted off-targets in matched bulk RNA-seq or scRNA-seq datasets from the same tissue (Supplementary Figures 9, 10, 11). We hope this evaluation pipeline will clarify to researchers they can evaluate whether predicted off-targets will appreciably affect results in their tissue of interest.

(3) Under-developed and non-essential software“The manuscript section on the software tool feels underdeveloped.”“Once the 10X Genomics corrects their gene panels according to this finding, the tool (OPT) will not be useful for most people. Still, it can be used by those who want to design de novo probes from scratch.”“Since the authors claim that OPT is intended for community use, the paper should provide a clear, step-by-step user guide, such as Jupyter tutorial, ideally as supplementary material.”

We agree with the reviewers that the description of the software tool itself is relatively concise. This is intentional, as the primary goal of this manuscript is not to introduce a standalone software framework, but rather to use the tool as a means to characterize and quantify off-target probe binding and its potential downstream impact on spatial gene expression analyses. Accordingly, our emphasis is placed on the biological and analytical insights enabled by this approach, rather than on extensive software tool details. To support potential users, we have now included additional software documented with an example Python notebook demonstrating how it can be applied to any probe panels in the GitHub repository: https://github.com/JEFworks-Lab/off-target-probe-tracker/blob/main/example.ipynb

Likewise, the primary goal of this manuscript is not to suggest that a specific vendor’s probe panels are flawed, but rather to demonstrate that off-target probe binding is a general and underappreciated phenomenon that can occur in some probe-based spatial transcriptomics platforms to meaningfully impact downstream analyses and biological interpretation.

OPT was developed as a framework to identify potential off-target probe interactions based on sequence homology. In practice, OPT can serve as a post hoc tool that allows researchers to assess whether predicted off-target interactions may exist in a given panel and to account for these possibilities when interpreting spatial expression patterns, even when panels have been developed by the many probe designing methods now highlighted in the revised manuscript. Given the complexity of probe design and hybridization behavior, we believe that explicitly identifying and reporting potential off-targets remains valuable for downstream data interpretation, cross platform comparisons, and reproducibility. Thus, OPT is intended to complement existing probe design strategies and vendor efforts, rather than replace them, by providing researchers with additional context to interpret their data more accurately.

In our revision, we have therefore elaborated on this in the discussion, reiterated here for convenience: “Although we focus here on the 10x Genomics Xenium technology, we do not exclude the possibility that off-target binding may similarly affect other probe-based gene detection approaches from other commercial vendors. Any technology that relies on hybridization-based detection is inherently susceptible to off-target probe binding when sequence similarity exists. Further, hybridization-based detection often inherently involves a trade-off between sensitivity and specificity. Given these inherent technological limitations, we therefore emphasize the importance of transparency through sharing probe sequences. However, many companies do not release the probe sequences used in their assays, limiting the consumer’s ability to fully interpret their results as well as the community’s ability to effectively characterize and benchmark performance variation across platforms. Therefore, we strongly recommend that companies publish probe sequences for pre-designed panels and likewise that researchers using these technologies should obtain and publish probe sequences used in their studies to support transparent and reproducible science. “

**Recommendations for the authors:**

“The paper only describes evidence of the off-target effect based on perfect sequence homology, although the tool (OPT) provides an option to find additional "potential" off-targets that allow mismatches. It would be very nice if the authors could additionally provide at least one example of off-target binding with at least one mismatch.”

We thank the reviewer for the opportunity to clarify this point. In addition to analyses based on perfect sequence homology, we examined predicted off-target binding when allowing mismatches at the terminal ends of probe sequences. This analysis is presented in the Results section titled “OPT results when allowing mismatches at the terminal ends of the probe sequences identifies additional off-target candidates.”

In this revision, we now allowed a 10bp padding on either end of the 40bp probe sequence, permitting imperfect sequence matching at the terminal regions. Under these conditions, OPT identified additional off-target candidates, including *TUBB2B* and *ACTG2*, which we highlight as representative examples (Supplementary 7,8). We further demonstrate how these predicted off-target interactions impact gene expression concordance by comparing Xenium measurements with both Visium and scRNA-seq data, showing measurable changes in cross-platform agreement. Together, these results illustrate that allowing mismatches reveals biologically relevant off-target effects beyond those captured by perfect sequence homology alone.

“Clarifications and updates for Figure 2A-BXenium offers a resolution of up to 200 nanometers with continuous readout, without pixel gaps. However, the figures shown in Figure 2A-B appear pixelated - why is this the case? Could the authors clarify this discrepancy and, if possible, provide the raw feature intensity data for Xenium in the supplementary materials?Additionally, there appear to be no visible gaps in the Visium graphs. Could the authors update the figure panels to represent the true spot locations for Visium, to more accurately reflect the underlying data structure?”

We thank the reviewer for the opportunity to clarify these points. The goal of Figure 2A-B is to facilitate a direct visual comparison of gene expression patterns between the Visium and Xenium platforms. To enable this comparison, we aggregated the single-cell Xenium data into spatial patches matching the effective resolution of Visium spots (55x55µm). Similarly, Visium spots were rendered as patches to produce a more continuous visual representation. As a result of this aggregation and visualization choice, the Xenium expression plots appear pixelated despite Xenium’s native subcellular resolution (up to ~200 nm with continuous readout). We have clarified this processing and visualization step in the Methods to avoid confusion.

With respect to the Visium expression plots, the lack of gaps is also a consequence of rendering each spot as a filled patch rather than plotting traditional Visium spots. This was done intentionally to maintain visual consistency with the aggregated Xenium data and to emphasize spatial concordance rather than the underlying sampling geometry. We have now explicitly stated this design choice to improve clarity.

“I found the format of the manuscript to be at times confusing and perhaps a bit of an odd fit for a general interest journal. A significant portion of the manuscript is spent critiquing a specific publication, "High resolution mapping of the tumor microenvironment using integrated single-cell, spatial and in situ analysis" published by Janesick et al. (of 10x Genomics, Inc) in Nature Communications in 2023. This content would seem more appropriate as a Comment submitted to Nature Communications, potentially to be accompanied by a response from the authors of Janesick et al. at 10x.”

I would like to address this important point as the corresponding author who takes primary responsibility for the unconventional decision to submit this manuscript to eLife as opposed to as a commentary suggested by the reviewer.

Consistent with the reviewer, I did initially consider submitting this as a Matters Arising to Nature Communications. However, after consultation with other senior colleagues and co-authors, I decided to forgo this route on the basis that the information provided in a Matters Arising must be kept confidential. I was concerned that this would lead to long, drawn-out private exchanges. As we note in the manuscript, the Xenium platform's widespread use and high cost imposed a certain urgency that I believed warranted open and rapid dissemination.

Therefore, we submitted to eLife with the hope that eLife’s unique continuous post-publication public peer review process will enable the rapid dissemination of these important financially-sensitive insights while permitting constructive criticisms from both industry and academic expert reviewers to be openly considered by all readers.